# Evaluation of potential sources of a priori ozone profiles for TEMPO tropospheric ozone retrievals

Matthew S. Johnson[1], Xiong Liu[2], Peter Zoogman[2,*], John Sullivan[3], Michael J. Newchurch[4], Shi Kuang[5], Thierry Leblanc[6], Thomas McGee[3]

[1]Earth Science Division, NASA Ames Research Center, Moffett Field, CA, USA.
[2]Harvard-Smithsonian Center for Astrophysics, Cambridge, MA, USA.
[3]Atmospheric Chemistry and Dynamics Laboratory, NASA Goddard Space Flight Center, Greenbelt, Maryland, USA.
[4]Atmospheric Science Department, University of Alabama in Huntsville, Huntsville, AL, USA.
[5]Earth System Science Center, University of Alabama in Huntsville, Huntsville, AL, USA.
[6]Table Mountain Facility, California Institute of Technology, Wrightwood, CA, USA.
*also at Minerva Schools at KGI, San Francisco, CA, USA.

*Correspondence to*: Matthew S. Johnson (matthew.s.johnson@nasa.gov)

**Abstract.** Potential sources of a priori ozone ($O_3$) profiles for use in Tropospheric Emissions: Monitoring of Pollution (TEMPO) satellite tropospheric $O_3$ retrievals are evaluated with observations from multiple Tropospheric Ozone Lidar Network (TOLNet) systems in North America. An $O_3$ profile climatology (tropopause-based $O_3$ climatology (TB-Clim), currently proposed for use in the TEMPO $O_3$ retrieval algorithm) derived from ozonesonde observations and $O_3$ profiles from three separate models (operational Goddard Earth Observing System (GEOS-5) Forward Processing (FP) product, reanalysis product from Modern-Era Retrospective analysis for Research and Applications version 2 (MERRA2), and the GEOS-Chem chemical transport model (CTM)) were: 1) evaluated with TOLNet measurements on various temporal scales (seasonally, daily, hourly) and 2) implemented as a priori information in theoretical TEMPO tropospheric $O_3$ retrievals in order to determine how each a priori impacts the accuracy of retrieved tropospheric (0-10 km) and lowermost tropospheric (LMT, 0-2 km) $O_3$ columns. We found that all sources of a priori $O_3$ profiles evaluated in this study generally reproduced the vertical structure of summer-averaged observations. However, larger differences between the a priori profiles and lidar observations were calculated when evaluating inter-daily and diurnal variability of tropospheric $O_3$. The TB-Clim $O_3$ profile climatology was unable to replicate observed inter-daily and diurnal variability of $O_3$ while model products, in particular GEOS-Chem simulations, displayed more skill in reproducing these features. Due to the ability of models, primarily the CTM used in this study, on average to capture the inter-daily and diurnal variability of tropospheric and LMT $O_3$ columns, using a priori profiles from CTM simulations resulted in TEMPO retrievals with the best statistical comparison with lidar observations. Furthermore, important from an air quality perspective, when high LMT $O_3$ values were observed, using CTM a priori profiles resulted in TEMPO LMT $O_3$ retrievals with the least bias. The application of near-real-time (non-climatological) hourly/daily model predictions as the a priori profile in TEMPO $O_3$ retrievals will be best suited when applying this data to study air quality or event-based processes as the standard retrieval algorithm will still need to use a climatology product. Follow-on studies to this work are currently being conducted to investigate the application of different CTM-predicted $O_3$ climatology products in the standard TEMPO retrieval algorithm. Finally, similar methods to those used in this study can be easily applied by TEMPO data users to recalculate tropospheric $O_3$ profiles provided from the standard retrieval using a different source of a priori.

## 1 Introduction

Ozone ($O_3$) is an important atmospheric constituent for air quality as concentrations above natural levels can have detrimental health impacts (US EPA, 2006) and the United States (US) Environmental Protection Agency (EPA) enforces surface-level mixing ratios under the National Ambient Air Quality Standards (NAAQS). In 2015, the NAAQS for $O_3$ was reduced from prior levels of 75 parts per billion (ppb) to 70 ppb, requiring that 3-year averages of the annual fourth-highest daily maximum 8-hour mean mixing ratio must be $\leq$ 70 ppb (US EPA, 2015). Tropospheric and surface-level $O_3$ mixing ratios are controlled by a complex system of photo-chemical reactions involving numerous trace gas species (e.g., carbon monoxide (CO), methane, volatile organic compounds, and nitrogen oxides ($NO_x$ = nitric oxide and nitrogen dioxide ($NO + NO_2$)) emitted from anthropogenic and natural sources (Atkinson, 1990; Lelieveld and Dentener, 2000). Furthermore, a substantial portion of tropospheric $O_3$ is also contributed from the downward transport from the stratosphere, commonly referred to as stratosphere-to-troposphere exchange (STE) (e.g., Stohl et al., 2003; Lin et al., 2015; Langford et al., 2017). Due to the complex chemistry and vertical/horizontal transport processes controlling $O_3$ mixing ratios, and the continued reduction of NAAQS levels, it is increasingly important to improve the ability to monitor/study tropospheric and surface-level $O_3$.

The monitoring of air quality in North America is typically conducted using ground-based in situ measurement networks. However, in recent years, observations of tropospheric $O_3$ and precursor gases (e.g., CO, $NO_2$, formaldehyde (HCHO)) have been made from space-borne platforms which have led to the better understanding of the tropospheric $O_3$ budget (Sauvage et al., 2007; Martin, 2008; Duncan et al., 2014). Total column (stratosphere + troposphere) $O_3$ has been routinely measured by numerous space-based sensors since the launch of systems such as the Total Ozone Mapping Spectrometer (TOMS) in 1978. Tropospheric column $O_3$ has been derived from total column retrievals using strategies such as residual-based approaches which subtract the stratospheric column $O_3$ from total $O_3$ (Fishman et al., 2008 and references therein). Tropospheric $O_3$ profiles have also been directly retrieved from hyperspectral Ultraviolet (UV) (e.g., Liu et al., 2005, 2010) and Thermal Infrared (TIR) (e.g., Bowman et al., 2006) measurements. Currently, sensors measuring tropospheric $O_3$, such as those using UV measurements from the Ozone Monitoring Instrument (OMI) and TIR measurements from the Tropospheric Emission Spectrometer (TES) (Beer, 2006), are from low earth orbit (LEO). While LEO provides global coverage, the observation of tropospheric $O_3$ is limited by coarse spatial resolution, limited temporal frequency (once or twice per day), and inadequate sensitivity to lower tropospheric and planetary boundary layer (PBL) $O_3$ (Fishman et al., 2008; Natraj et al., 2011). These limitations restrict the ability to apply these space-borne observations in air quality policy and monitoring.

The Tropospheric Emissions: Monitoring of Pollution (TEMPO) instrument, which will be launched between 2019-2021 to geostationary orbit (GEO), is designed to address some of the limitations of current $O_3$ remote-sensing instruments (Chance et al., 2013; Zoogman et al., 2017). TEMPO will provide critical measurements such as vertical profiles of $O_3$, total column $O_3$, $NO_2$, sulfur dioxide, HCHO, glyoxal, and aerosol/cloud parameters over North America. These data products will be provided at temporal resolutions as high as hourly and at a native spatial resolution of ~2.1 × 4.4 km$^2$ (at the center of the field of regard) except at the required spatial resolution of 8.4 × 4.4 km$^2$ for the $O_3$ profile product (four pixels combined to increase signal to noise ratios and reduce computational resources). TEMPO's domain will encompass the region of North America from Mexico City to the Canadian oil

sands and from the Atlantic to the Pacific Ocean. TEMPO will have increased sensitivity to lower tropospheric $O_3$
compared to past/current satellite data by combining measurements from both UV (290-345 nm) and visible (VIS,
540-650 nm) wavelengths (Natraj et al., 2011; Chance et al., 2013; Zoogman et al., 2017). The operational TEMPO
$O_3$ product will provide vertical profiles and partial $O_3$ columns at ~24-30 layers from the surface to ~60 km above
ground level (agl). This product will also include total, stratospheric, tropospheric, and a 0-2 km agl $O_3$ columns.
TEMPO's high spatial and temporal resolution measurements, including the 0-2 km $O_3$ column, will provide a wealth
of information to be used in air quality monitoring and research.

Vertical $O_3$ profile retrievals from TEMPO will be based on the Smithsonian Astrophysical Observatory

(SAO) $O_3$ profile algorithm which was developed for use in the Global Ozone Monitoring Experiment (GOME) (Liu
et al., 2005), OMI (Liu et al., 2010), GOME-2 (Cai et al., 2012), and the Ozone Mapping and Profiler Suite (Bak et
al., 2017). Currently, the SAO $O_3$ retrieval algorithm for TEMPO has proposed to apply the tropopause-based $O_3$
climatology (TB-Clim) developed in Bak et al. (2013) as the a priori profiles (Zoogman et al., 2017), which was
demonstrated to improve OMI $O_3$ retrievals near the tropopause compared to calculations using the Labow-Logan-
McPeters (LLM) $O_3$ climatology (a priori used for OMI) (McPeters et al., 2007). During this work, we evaluate the
representativeness of the vertical $O_3$ profiles from TB-Clim. Additionally, we evaluate simulated near-real-time (NRT,
term used in this study to identify non-climatological/time-specific products) $O_3$ profiles from an operational data
assimilation model product (National Aeronautics and Space Administration (NASA) Global Modeling and
Assimilation Office (GMAO) Goddard Earth Observing System (GEOS-5) Forward Processing (FP)), a reanalysis
data product (NASA GMAO Modern-Era Retrospective analysis for Research and Applications version 2
(MERRA2)), and a chemical transport model (CTM) (GEOS-Chem). The climatology and model $O_3$ profiles were
evaluated with ground-based lidar data from the Tropospheric Ozone Lidar Network (TOLNet) at various locations
of the US during the summer of 2014. This evaluation focused on the performance of each product compared to
summer-, daily-, and hourly-averaged lowermost tropospheric (LMT, 0-2 km) and tropospheric (0-10 km) $O_3$ columns.
Furthermore, based on past studies demonstrating the importance of a priori profiles in trace gas satellite retrievals
(Martin et al., 2002; Luo et al., 2007; Kulawik et al., 2008; Zhang et al., 2010, Bak et al., 2013), we evaluated the
effectiveness of using the TB-Clim and model products as a priori in the TEMPO $O_3$ profile algorithm.

This paper is organized as follows. Section 2 describes the tropospheric lidar $O_3$ measurements, TB-Clim and

model products, theoretical TEMPO retrievals, and data evaluation techniques applied during this study. Section 3
provides the results of the comparison of the TB-Clim and modeled a priori profile products with TOLNet observations
and the impact of each product, when applied as a priori, on TEMPO tropospheric $O_3$ profile retrievals. Finally, Sect.
4 concludes this study.
**2 Data and methods**
**2.1 TOLNet**
TOLNet provides Differential Absorption Lidar (DIAL)-derived vertically-resolved $O_3$ mixing ratios at 6 different
locations of North America (http://www-air.larc.nasa.gov/missions/TOLNet/). TOLNet data have been used
extensively in atmospheric chemistry research on topics such as STE, air pollution transport, nocturnal $O_3$
enhancements, PBL pollution entrainment, source attribution of $O_3$ lamina, and the impact of wildfire and lightning
$NO_x$ on tropospheric $O_3$ (e.g., Kuang et al., 2011; Sullivan et al., 2015a, 2016, Johnson et al., 2016; Granados-Muñoz
et al., 2017; Langford et al., 2017). Uncertainty in TOLNet $O_3$ measurements due to systematic error are approximately
4-5% for all instruments at all altitudes. Precision will vary from 0% to > 20% and is dependent on individual
instrument characteristics, time of day, and temporal and vertical averaging (precision typically degrades with height
for altitudes above 8-10 km) (Kuang et al., 2013; Sullivan et al., 2015b; Leblanc et al., 2016). Since TOLNet
observations used during this study are hourly-averaged and typically below 10 km agl, overall uncertainty can be
assumed to be ≤ 10%. TOLNet data were applied in this study to evaluate the TB-Clim and model-predicted profiles
which could potentially be used as TEMPO a priori information. Furthermore, theoretical TEMPO $O_3$ retrievals in the
troposphere and LMT were calculated using the climatology/model profiles as a priori with TOLNet data representing
the "true" atmospheric $O_3$ profiles (see Sect. 2.2).
During this study, vertical $O_3$ profiles from 3 separate TOLNet sites during the summer (July-August) of
2014 were applied. Figure 1 shows the location of the Goddard Space Flight Center (GSFC) TROPospheric OZone
(TROPOZ), Jet Propulsion Laboratory (JPL) Table Mountain Facility (TMF), and the University of Alabama in
Huntsville (UAH) Rocket-city $O_3$ Quality Evaluation in the Troposphere (RO3QET) TOLNet systems which provided
the observations used during this work. These 3 sites were selected due to data availability (http://www-
air.larc.nasa.gov/missions/TOLNet/data.html) and to represent differing parts of North America, which will be
observed by TEMPO, with varying topography, meteorology, and atmospheric chemistry conditions (overview
information for each station is presented in Table 1). The RO3QET system is located in the southeast US where the
air quality is impacted by both anthropogenic and natural emission sources, complex chemistry, and multiple transport
pathways (e.g., Hidy et al., 2014; Johnson et al., 2016; Kuang et al., 2017). During the summer of 2014 this lidar
system measured $O_3$ profiles from the surface to ~5 km agl during the daytime hours. The TROPOZ system, which is
typically operated at NASA GSFC, was remotely stationed in Fort Collins, Colorado to support the Deriving
Information on Surface Conditions from COlumn and VERtically Resolved Observations Relevant to Air Quality
(DISCOVER-AQ) Colorado and Front Range Air Pollution and Photochemistry Éxperiment (FRAPPÉ) field
campaigns between July-August 2014. The TROPOZ system was arranged to take daytime observations of $O_3$ profiles
in the intermountain west region of the US alongside the frontal range of the Rocky Mountains. The air quality of this
location is impacted by large anthropogenic emission sources, complex local transport, and common STE events (e.g.,
Sullivan et al., 2015a, 2016; Vu et al., 2016). Finally, the TOLNet system at the JPL TMF is representative of the
western US and remote high-elevation locations. This location has $O_3$ profiles largely controlled by long-range
transport and STEs typical of remote high-elevation locations in the US (e.g., Granados-Muñoz and Leblanc, 2016;
Granados-Muñoz et al., 2017). During the summer of 2014, the JPL TMF lidar only conducted measurements during
the nighttime hours and therefore will only be used for daily-averaged comparisons to TB-Clim and model predictions.


**2.2 TEMPO O$_3$ profile retrieval**

TEMPO will adapt the current SAO OMI UV-only O$_3$ profile algorithm (Liu et al., 2010) to derive O$_3$ profiles from joint UV+VIS measurements based on the optimal estimation technique (Rodgers, 2000). Partial O$_3$ columns at different altitudes, along with other retrieved variables, are iteratively derived by simultaneously minimizing the differences between measured and simulated radiances and between the retrieved and a priori state vectors. For this study, we use the linear estimate approach to perform theoretical TEMPO retrievals and evaluate the impact of a priori profiles on these retrievals. This linear estimation approach is a good first-order approximation of non-linear satellite retrievals and has been used in numerous research studies (e.g., Bowman et al., 2002; Worden et al., 2007; Kulawik et al., 2006, 2008; Zhang et al., 2010; Natraj et al., 2011; Zoogman et al., 2014). In this approach, shown in Eq. (1), the retrieved O$_3$ profile ($X_r$) is derived as:

$$X_r = X_a + A(X_t - X_a) + G\varepsilon, \tag{1}$$

where $X_a$ is the a priori O$_3$ profile, $A$ is the averaging kernel (AK) matrix, $X_t$ is the true O$_3$ profile, $G$ is the gain matrix, and $\varepsilon$ is the measurement noise. The last term on the right represents the retrieval precision. During this study, no measurement noise/error is taken into account. The error component adds measurement noise to the linear retrievals, however, neglecting this term does not affect the inter-comparison of the impact of individual a priori sources on TEMPO retrieved tropospheric O$_3$. The linear estimation approach represented in Eq. (1) assumes no worse than moderate non-linearity between the retrieved and true state (Rodgers, 2000). Furthermore, during this study pre-computed AKs (see Sect. 2.2.1) are used with multiple different a priori profiles to determine the impact of varying O$_3$ a priori sources on TEMPO retrieved tropospheric O$_3$. In order to apply Eq. (1), with pre-computed AKs and varying a priori profiles, it must be assumed that there is only moderate non-linearity between the retrievals. The assumption of linearity taken during this study is validated in Appendix A.

**2.2.1 TEMPO averaging kernels**

The UV+VIS AKs applied during this study were pre-computed during TEMPO retrieval sensitivity studies that played a key role in determining the instrument requirements and verification of the retrieval performance (Zoogman et al., 2017). The production of these AKs involved: 1) radiative transfer model simulations of TEMPO radiance spectra and weighting functions calculated using GEOS-Chem vertical profiles and 2) retrieval AKs and errors calculated from the weighting functions, TB-Clim a priori error covariance matrix, and measurement random-noise error covariance matrix estimated using the TEMPO signal to noise ratio model. To represent TEMPO hourly measurements throughout the year, the retrieval sensitivity calculation was performed hourly for 12 days (15[th] day of each month) over the TEMPO domain at a spatial resolution of 2.0°×2.5° (latitude × longitude) using hourly GEOS-Chem model fields for the year 2007. Here we present a basic overview of the methods used in the TEMPO retrieval sensitivity studies and those to produce the pre-computed AKs, however, for detailed information about the methods and input variables see Zoogman et al. (2017). To represent atmospheric conditions retrieved by the TEMPO sensor in the retrieval sensitivity studies, GEOS-Chem trace gas and aerosol fields and GEOS-5 meteorological data were applied over the TEMPO field of regard. Viewing geometry, radiance spectra, and weighting functions (calculated

with the VLIDORT radiative transfer model at a spectral resolution of 0.6 nm and intervals of 0.2 nm for solar zenith
angles ≤ 80°) with respect to aerosols and trace gases were all calculated based on TEMPO specifications. Surface
albedo values were taken from the GOME albedo database. Optimal estimation was applied to conduct the TEMPO
retrieval sensitivity studies and $O_3$ profile retrievals. During these retrieval sensitivity studies, the AK values were
calculated using Eq. (2):
$AK = \frac{\partial \hat{x}}{\partial x_t} = \hat{S} K^T S_{yn}^{-1} K = GK$                                                                 (2)
where $\hat{x}$ is the retrieved state vector, $x_t$ is the unknown true state vector, $\hat{S}$ is the solution error covariance matrix, $K$
is the weighting function matrix ($K = \frac{\partial y}{\partial x}$, $y$ is the observed radiances), and $S_{yn}$ is the measurement random-noise
error covariance matrix.

During this study, we used the UV+VIS $O_3$ retrieval AKs corresponding to the month and location of TOLNet

systems representative of near clear-sky conditions. Figure 2 shows an example of the UV+VIS AK matrix at the
UAH RO3QET site for 20 UTC in August. The enhanced sensitivity of TEMPO retrievals in the lower troposphere,
in particular the lowest ~2 km, is demonstrated by the large values of $A$ (normalized to 1 km, degrees of freedom
(DFS) per km) in Fig. 2 (> 0.20). When including VIS with UV wavelengths, $O_3$ retrievals can be greater than a factor
of 2 more sensitive in the first 2 km of the troposphere in comparison to just using UV wavelengths. This is particularly
important as accurate $O_3$ observations between 0-2 km agl is a key requirement of TEMPO to be a sufficient data
source for air quality research/monitoring (Zoogman et al., 2017).
**2.2.2 TB-Clim**
During this study, TB-Clim is evaluated with observations to determine the ability of these profiles to represent the
spatio-temporal variability of tropospheric $O_3$ in North America. A detailed description of the data and procedures
used to derive TB-Clim can be found in Bak et al. (2013). The climatology provides monthly-averaged $O_3$ profiles
with 1 km vertical resolution relative to the tropopause in 18 10°-latitude bins (Bak et al., 2013). During this study,
hourly TB-Clim $O_3$ profiles were derived by applying hourly-averaged GEOS-5 FP tropopause heights. Figure 3
illustrates the monthly-averaged vertical structure of TB-Clim that will be evaluated at the RO3QET, TROPOZ, and
JPL TMF system locations representative of various regions of the US in July-August 2014. At the location of the
RO3QET system (Fig. 3, green line), $O_3$ values are ~55 ppb near the surface during July and August and steadily
increase to ~95 ppb at 10 km. For the location of the TROPOZ system (Fig. 3, black line), $O_3$ values are ~40-45 ppb
near the surface and increase to ~80 ppb at 10 km. Finally, at the location of the JPL TMF lidar system (Fig. 3, red
line), $O_3$ values are ~50-55 ppb near the surface and increase to 80-95 ppb at 10 km.
**2.3 Simulated $O_3$ profile data**
Satellite $O_3$ retrieval algorithms typically apply climatologies derived from observational data (i.e., ozonesondes) as
a priori information (Liu et al., 2005, 2010; Cai et al., 2012). However, some satellites, such as TES operational
retrievals, apply climatological $O_3$ profiles from global CTMs as a priori information (Worden et al., 2007). During
this work, we evaluate NRT $O_3$ profile information from an operational data assimilation model (GEOS-5 FP),
reanalysis model (MERRA2), and a CTM (GEOS-Chem) using TOLNet data and investigate how these model
products impact theoretical TEMPO $O_3$ retrievals when applied as a priori information. Due to numerous reasons, the
standard TEMPO $O_3$ profile algorithm will need to apply an hourly-resolved monthly mean climatology, however, we
evaluated NRT model data here as TEMPO data users can simply apply the outputs from the standard retrieval to
recalculate the tropospheric $O_3$ vertical profiles using a different source of a priori. These simulated products were
selected to represent model predictions of $O_3$ with highly varying complexity in atmospheric chemistry calculations,
emissions information, data assimilation techniques, and spatial resolution.
**2.3.1 GEOS-5 FP and MERRA2**
The GEOS-5 atmospheric general circulation model (AGCM) and data assimilation system (DAS) is a product of the
GMAO and is described in Rienecker et al. (2008) with most recent updates presented in Molod et al. (2012). Aerosol
and trace gases are transported in the GEOS-5 AGCM using a finite-volume dynamics scheme implemented with
various physics packages (Putman and Lin, 2007; Bacmeister et al., 2006) and turbulently mixed using the Lock et al.
(2000) PBL scheme. The GEOS-5 AGCM ADS assimilates roughly $2\times10^6$ observations for each analysis using the
Gridpoint Statistical Interpolation (GSI) three-dimensional variational (3DVar) analysis technique (Wu et al., 2002).
A product from the GEOS-5 AGCM is the operationally provided GEOS-5 FP data which offers NRT DAS predictions
(typically within 24 hours) of $O_3$ vertical profiles at a 0.25°×0.3125° spatial resolution and 72 vertical levels.
Additionally, we apply MERRA2 reanalysis $O_3$ profiles which are also produced using the GEOS-5 AGCM (Molod
et al., 2012) and provided at a 0.50°×0.667° spatial resolution and 72 vertical levels. Both GEOS-5 FP and MERRA2
$O_3$ vertical profiles are driven by the assimilation of OMI and Microwave Limb Sounder (MLS) satellite data.
Predictions of $O_3$ from these products are most trusted in the upper troposphere and stratosphere due to OMI and MLS
having limited sensitivity in the lower troposphere (e.g., Wargan et al., 2015; Ott et al., 2016). The work by Wargan
et al. (2015) showed that due to highly simplified atmospheric chemistry and lack of surface emissions in the GEOS-
5 AGCM, $O_3$ predictions in the middle to lower troposphere tend to be biased. However, during this work these 3
hour-averaged products are applied to understand how NRT DAS and reanalysis models could be used as a priori
information in TEMPO $O_3$ retrievals.
**2.3.2 GEOS-Chem**
GEOS-Chem (v9-02) was applied in this work as a proxy to determine how a full CTM or air quality model could
potentially be used as a priori information in TEMPO $O_3$ retrievals. The purpose of this work is not to evaluate the
performance of the GEOS-Chem model, or to suggest GEOS-Chem as the only model to provide a priori information
for TEMPO, but to simply evaluate how CTM predictions impact the accuracy of theoretical TEMPO $O_3$ retrievals.
The CTM is driven by GEOS-5 FP meteorological data in a nested regional mode for July and August 2014, after a
2-month spin-up period, at a 0.25°×0.3125° spatial resolution and 47 hybrid terrain following vertical levels for the
North American domain (130°-60°W, 9.75°-60°N). GEOS-Chem includes detailed $O_3$-$NO_x$-hydrocarbon-aerosol
chemistry coupled to $H_2SO_4$-$HNO_3$-$NH_3$ aerosol thermodynamics (Bey et al., 2001). Furthermore, aerosol and trace
gas transport are calculated using the TPCORE parameterization (Lin and Rood, 1996) and dry and wet deposition
(Wang et al., 1998; Amos et al., 2012) is simulated on a 10-minute time-step. A detailed description of the version of
GEOS-Chem, and emission inventories, applied during this study can be found in Johnson et al. (2016).

**2.4 Data evaluation**

The evaluation of TB-Clim and model $O_3$ profiles was done for summer-, daytime- (6am - 6pm local time), and hourly-
averages at the RO3QET and TROPOZ system locations during July and August 2014. Due to the hours of operation,
the evaluation at the JPL TMF lidar location was not conducted for hourly-averages and is only applied for summer-
and daily-averages. To determine the ability of a NRT DAS, reanalysis, and CTM model to replicate TOLNet-
observed $O_3$, GEOS-5 FP, MERRA2, and GEOS-Chem data was evaluated simultaneously with TB-Clim. For all
evaluation and inter-comparisons, TB-Clim, model data, TOLNet observations, and TEMPO calculations were hourly-
averaged and averaged/interpolated to the vertical grid of the TEMPO AKs during all times/locations when/where
TOLNet measurements were obtained. TB-Clim and model data used as a priori, and resulting $X_r$ calculations, were
evaluated using statistical parameters (correlation (R), bias, bias standard deviation (1σ), mean normalized bias
(MNB), root mean squared error (RMSE)) and time-series analysis for tropospheric (0-10 km, 0-5 km for RO3QET)
and LMT (0-2 km) columns. Tropospheric column values are considered to extend from the surface to 10 km in this
study based on the fact that TOLNet systems typically only measured to ~10 km agl.

**3 Results**

**3.1 Evaluation of TB-Clim and model-predicted tropospheric $O_3$ profiles**

In terms of summertime-averaged tropospheric $O_3$ profiles, TB-Clim and the GEOS-5 FP, MERRA2, and GEOS-
Chem models could generally replicate the vertical structure of tropospheric $O_3$ measured by TOLNet lidars. However,
the evaluation of these products as a priori in TEMPO $O_3$ retrievals at a seasonal/monthly average is insufficient as
TEMPO will provide hourly, high spatial resolution, tropospheric and LMT $O_3$ values. Therefore, in the following
sections we evaluate these products for daily- and hourly-averages to focus on inter-daily and diurnal variability.

**3.1.1 Daily-averaged tropospheric $O_3$ profiles**

This section focuses on evaluating the ability of TB-Clim and the GEOS-5 FP, MERRA2, and GEOS-Chem models
to reproduce observed daily variability of $O_3$ in the troposphere and near the surface. Figure 4 shows the daily-averaged
tropospheric and LMT $O_3$ columns from TB-Clim and models compared to that observed by TOLNet at all 3 sites
with comparison statistics displayed in Table 2. Some slight inter-daily variability can be seen in TB-Clim tropospheric
$O_3$ due to varying time-dependent tropopause heights, however, the variability in LMT values is mostly due to only
sampling values in the vertical layers and times when TOLNet observations were obtained (vertical layers of TOLNet
observations varied between hours and days). Due to the zonal and monthly mean nature of TB-Clim, this dataset is
unable to replicate inter-daily $O_3$ observations consistently displaying low and negative correlation values with daily
TOLNet observations in the troposphere (R range between -0.09 and -0.35) and near the surface (R range between -
0.15 and -0.68). The models demonstrate a better ability to replicate the daily variability of observed tropospheric $O_3$
at the TOLNet system locations. Overall, CTM predictions from GEOS-Chem was the only source of $O_3$ profiles
which consistently displayed moderate to high positive correlation (all R values > 0.47) compared to all TOLNet
observations in the troposphere and near the surface. This result is not overly surprising as a full CTM includes aspects
necessary to reproduce the spatio-temporal tropospheric $O_3$ variability occurring in nature such as data-assimilated
meteorological fields, comprehensive atmospheric chemistry mechanisms, and state-of-the-art trace gas and aerosol
emissions data.
Figure 4a, b shows larger variability of daily-averaged LMT $O_3$ (44 to 68 ppb) from the RO3QET system
than that in the tropospheric column (48 to 64 ppb). From Table 2 it can be seen that TB-Clim was generally high
compared to lidar-measured tropospheric $O_3$ mixing ratios (average bias = 3.7 ppb) with large bias standard deviations
and RMSE values (> 6 ppb). MERRA2 displayed good agreement in tropospheric $O_3$ (negative bias ~0.7 ppb) while
GEOS-5 FP and GEOS-Chem resulted in moderate high biases (average bias 2.8 and 1.7 ppb, respectively). GEOS-
Chem had moderate high biases but with smaller bias standard deviation and RMSE values (< 4.5 ppb) in comparison
to the other products due to the ability to better capture inter-daily tropospheric $O_3$ variability (R = 0.61). LMT $O_3$
observations by the RO3QET lidar were best replicated by the CTM product resulting in the smallest average bias (-
1.3 ppb) and bias standard deviation and RMSE values (4.4 ppb) compared to the other products. MERRA2 was
consistently low compared to LMT $O_3$ observations (bias = -4.9 ppb) while TB-Clim and GEOS-5 FP resulted in
moderate biases (2.9 and -2.9 ppb, respectively) with all of these products having large bias standard deviations and
RMSE (≥ 8.0 ppb).
At the TROPOZ system location, large variability in tropospheric (47 to 83 ppb) and LMT $O_3$ values (41 to
73 ppb) was observed. From Fig. 4c, d and Table 2 it can be seen that TB-Clim is unable to replicate the inter-daily
tropospheric $O_3$ variability and is generally higher in comparison to observations with large bias standard deviations
(bias ± standard deviation = 2.2 ± 9.7 ppb). GEOS-Chem best replicates the daily variability of tropospheric $O_3$ with
the largest correlation (R = 0.82) and small average bias and standard deviations (2.4 ± 6.0 ppb). GEOS-5 FP and
MERRA2 data displayed low positive correlations (R < 0.40) and larger average biases and standard deviations of 3.3
± 10.0 and -4.6 ± 9.1 ppb, respectively. In comparison to TROPOZ LMT $O_3$ observations, TB-Clim and all model
products displayed large negative biases. The TB-Clim product resulted in the largest negative biases and bias standard
deviations compared to LMT $O_3$ observations (-11.1 ± 7.5 ppb) and model products displayed smaller biases and
standard deviations. GEOS-5 FP data displayed the lowest average bias (-4.4 ppb) compared to TROPOZ
observations, however, was unable to replicate the inter-daily variability of LMT $O_3$ (R = -0.09) resulting in large bias
standard deviations (7.3 ppb). Overall, GEOS-Chem was the only product which was able to capture the inter-daily
variability of LMT $O_3$ (R = 0.47) resulting in moderate low biases and the lowest bias standard deviation (-6.7 ± 6.2
ppb).
Figure 4e, f illustrates that large inter-daily variability of tropospheric (46 to 129 ppb) and LMT (35 to 76
ppb) column $O_3$ was observed at the JPL TMF site during the summer of 2014. This figure and Table 2 shows that
TB-Clim is able to represent the average magnitude of tropospheric $O_3$ (bias = 0.3 ppb) but with large bias standard
deviation and RMSE values (>18 ppb) due to the inability to replicate observed inter-daily variability (R = -0.35). The
GEOS-Chem model also captures the average magnitude of tropospheric $O_3$ (bias = -0.5 ppb) but with smaller bias
standard deviations (14.6 ppb) compared to TB-Clim due to the ability to better replicate the inter-daily availability
(R = 0.72). GEOS-5 FP and MERRA2 demonstrated negative biases compared to JPL TMF lidar observed
tropospheric $O_3$ (-5.0 and -10.6 ppb, respectively) with relatively low bias standard deviations (~13-14 ppb) compared
to the other products. The large RMSE values for all products is due to the very large variability in daily-averaged $O_3$
observations which was not well captured by all products. Near the surface, the GEOS-Chem model clearly best
captures the variability of daily-averaged LMT $O_3$ indicated by the smallest bias and standard deviations (0.9 ± 10.4
ppb) and RMSE (~10.25 ppb) values.
**3.1.2 Diurnal cycle of tropospheric $O_3$ profiles**
TEMPO retrievals will produce hourly tropospheric and LMT $O_3$ values each day for the entire North America
domain. Therefore, this section focuses on evaluating the ability of TB-Clim and the GEOS-5 FP, MERRA2, and
GEOS-Chem models to reproduce the observed diurnal variability of $O_3$ measured at the RO3QET and TROPOZ
system locations in the troposphere and near the surface. Figure 5 shows the average diurnal time-series of hourly-
averaged tropospheric and LMT $O_3$ (from all days of observation) from the $O_3$ climatology and models compared to
that observed during the summer of 2014 (statistics displayed in Table 3).
Figure 5a, b shows that larger diurnal variability of $O_3$ was observed for LMT values (48 to 59 ppb) compared
to tropospheric values (55 to 60 ppb) at the RO3QET lidar location. All the sources of $O_3$ profiles evaluated here,
excluding the CTM predictions, demonstrate very little diurnal variation in tropospheric and LMT $O_3$ at the RO3QET
lidar location. The GEOS-Chem model was the only product able to replicate the diurnal variability of observed
tropospheric $O_3$ (R = 0.68). MERRA2 resulted in the lowest bias (-1.2 ppb), GEOS-5 FP and GEOS-Chem displayed
modest biases (~2.0-2.5 ppb), and TB-Clim had the largest bias (3.5 ppb) compared to RO3QET tropospheric $O_3$ data.
Diurnal RO3QET LMT $O_3$ data was best replicated by CTM predictions resulting in the highest correlation (R = 0.76),
lowest bias and standard deviations (0.3 ± 2.6 ppb), and RMSE values (2.45 ppb). The TB-Clim product resulted in
modest biases compared to LMT $O_3$ data (1.9 ppb) while GEOS-5 FP and MERRA2 were consistently low (negative
bias > 3.0 ppb).
Figure 5c, d shows the diurnal variability of $O_3$ that was observed for tropospheric and LMT column values
at the TROPOZ lidar location. In the troposphere, $O_3$ values varied between ~58 to 69 ppb with largest values
occurring in the afternoon. Larger diurnal variability was observed near the surface with LMT $O_3$ values ranging from
~56 to 75 ppb with largest values occurring between 21 and 05 UTC. GEOS-Chem data was the only product which
could replicate the diurnal variability of TROPOZ lidar tropospheric $O_3$ observations (R = 0.78). The TB-Clim, GEOS-
5 FP, and GEOS-Chem products demonstrate moderate high biases (2.2-3.3 ppb) compared to the observations while
MERRA2 was consistently low (bias = -5.1 ppb). For comparison of near-surface $O_3$ values (see Fig. 5d), none of the
products sufficiently captured the magnitude and degree of diurnal variability of LMT $O_3$ at the TROPOZ lidar
location. The TB-Clim product displayed a small positive correlation (R = 0.26) and large negative biases (-12.6 ppb),
bias standard deviation (6.9 ppb), and RMSE values (14.25 ppb). The GEOS-5 FP and GEOS-Chem models display
the lowest bias (negative bias between 7.5 ppb and 7.7 ppb), however, the CTM is more highly correlated (R = 0.92)
and resulted in lower bias standard deviations (4.8 ppb) and RMSE values (9.01 ppb). This indicates that while no
product reproduced the magnitude or degree of diurnal variability of near-surface $O_3$ observed by the TROPOZ lidar,
the GEOS-Chem CTM does the best job on average.

**3.2 Prior $O_3$ vertical profile impact on TEMPO retrievals**

This section focuses on how the TB-Clim, GEOS-5 FP, MERRA2, and GEOS-Chem $O_3$ profiles impact theoretical
TEMPO tropospheric $O_3$ profile retrievals when applied as the a priori information in Eq. (1). The evaluation is focused
on how different sources of a priori impacted the overall accuracy of TEMPO tropospheric $O_3$ retrievals and the ability
to meet the required precision of tropospheric and LMT $O_3$ observations of 10 ppb (Zoogman et al., 2017). The
requirement for TEMPO tropospheric $O_3$ is that retrieval errors (root square sum of retrieval precision and smoothing
errors) or overall biases should be < 10 ppb, and, therefore, we quantify the number of occurrences when total error
or bias standard deviation/RMSE exceeds this 10 ppb limit. TEMPO will provide tropospheric and LMT $O_3$ at high
temporal resolution and therefore, $X_r$ values from Eq. (1), using the individual a priori sources, were evaluated on a
daily-averaged and diurnal cycle time scale.

**3.2.1 Tropospheric $O_3$ TEMPO retrievals**

Figure 6 shows the time-series of daily-averaged tropospheric and LMT $X_r$ column values and bias calculations when
using TB-Clim and model data as a priori information when compared to observed $O_3$ at all 3 TOLNet sites (statistics
in Table 4). When focusing on the accuracy of the theoretical TEMPO retrievals for tropospheric $X_r$ columns (left
column in Fig. 6), it can be seen that $X_r$ values using all a priori profiles: 1) are similar, 2) are highly correlated with
observations (see Table 4), and 3) compare well to observations with tropospheric $X_r$ values typically falling within
the 10 ppb bias requirement at all 3 TOLNet locations. From Table 4 it can be seen that daily-averaged tropospheric
column biases exceeded the 10 ppb level on 1 and 2 days when using TB-Clim/GEOS-5 FP and MERRA2 data,
respectively, as a priori when compared to TROPOZ observations, and for 1 day at the JPL TMF location when using
all $O_3$ products as a priori.

Table 4 illustrates that applying TB-Clim as the a priori resulted in the largest tropospheric column $X_r$ biases

and modest bias standard deviations ($1.4 \pm 2.3$ ppb) and the MERRA2 data led to the lowest overall bias and modest
bias standard deviation ($-0.2 \pm 2.5$ ppb) at the RO3QET lidar location. Using GEOS-Chem a priori profiles resulted
in modest biases and the lowest bias standard deviations ($1.0 \pm 2.0$ ppb) and RMSE values (2.17 ppb). At the TROPOZ
system site, the lowest tropospheric column $X_r$ biases and standard deviation were calculated when applying GEOS-
Chem as the a priori ($-0.5 \pm 2.7$ ppb). GEOS-5 FP data also resulted in low mean $X_r$ biases but the largest bias standard
deviations ($-0.6 \pm 4.8$ ppb) and MERRA2 data led to larger mean $X_r$ biases but lower bias standard deviations ($-2.2 \pm$
$4.4$ ppb). The use of TB-Clim resulted in modest mean bias and standard deviations ($-0.9 \pm 4.2$ ppb). Finally, at the
JPL TMF location all a priori profile sources resulted in average tropospheric column $X_r$ biases of < 1.0 ppb, excluding
MERRA2 (bias = -1.7 ppb), with similar bias standard deviations and RMSE values (ranging between 3.0 to 4.0 ppb).
Much larger daily variability of tropospheric $O_3$ was observed at the JPL TMF site compared to the other TOLNet
system locations and tropospheric column $X_r$ values from theoretical TEMPO retrievals successfully captured this
variability using all the sources of a priori information. These results suggest that TEMPO, using UV+VIS
wavelengths, will likely be able to accurately retrieve highly variable tropospheric column $O_3$ magnitudes regardless
of the a priori profile used.

**3.2.2 LMT $O_3$ TEMPO retrievals**

The third column of Fig. 6 shows that much larger differences in daily-averaged LMT column $X_r$ values were
calculated, compared to tropospheric $X_r$ values, when using different sources of a priori in Eq. (1). From this figure
and Table 4 it can be seen that LMT column $X_r$ values better capture the daily variability of near-surface $O_3$ compared
to the a priori profiles, however, noticeable differences in the statistical comparison of LMT column $X_r$ values using
different a priori sources are evident. It can be seen from this figure that at the RO3QET site, daily variability of near-
surface $O_3$ are clearly best captured by LMT $X_r$ values using GEOS-Chem CTM a priori profiles. While the TB-Clim
product resulted in LMT $X_r$ values with the smallest mean bias (0.2 ppb), it also led to large RMSE values (5.88 ppb)
and the largest bias standard deviations (6.1 ppb) (see Table 4). Table 4 illustrates that LMT column $X_r$ values
calculated using CTM a priori profiles had modest mean bias (-2.2 ppb) and the lowest bias standard deviations (2.5
ppb) and RMSE (3.26 ppb). Applying the GEOS-5 FP and MERRA2 model products as a priori profiles resulted in
the largest mean biases in LMT $X_r$ values (negative biases $\geq$ 3.4 ppb) along with largest RMSE values ($\geq$ 6.0 ppb).
From an air quality perspective, it is important to note that LMT column $X_r$ values using a priori data other than
GEOS-Chem are unable to replicate the larger surface $O_3$ values occurring in the southeast US (see Fig. 6). A few
LMT $O_3$ accuracy/precision requirement exceedances were calculated at the RO3QET lidar location using all a priori
products except for GEOS-Chem predictions. The ability of GEOS-Chem to best reproduce the magnitude of the daily
LMT $O_3$ variability resulted in LMT $X_r$ values with the smallest RMSE and bias standard deviations, no
accuracy/precision requirement exceedances, and the best ability to capture the range in daily observed $O_3$.

At the location of the TROPOZ lidar, it can be seen from Fig. 6 that LMT $X_r$ values, with the use of TB-

Clim a priori, are consistently underestimated in comparison to lidar observations. These LMT $X_r$ values have an
average negative bias of > 10.0 ppb and largest RMSE values (~13.0 ppb) resulting in 10 days with accuracy/precision
requirement exceedances (see Table 4). These large errors are because the a priori profiles provided by TB-Clim are
not able to replicate the highly variable vertical $O_3$ profiles observed at the TROPOZ lidar location. The GEOS-5 FP,
MERRA2, and GEOS-Chem models were better able to replicate these highly variable vertical $O_3$ profiles providing
a priori information more accurately representing $O_3$ in the intermountain west region of the US. This better
representation from model data resulted in LMT $X_r$ values with lower negative mean biases (< 6.5 ppb) and smaller
RMSE values (< 9.0 ppb) and bias standard deviations (< 6.5 ppb), and also fewer accuracy/precision requirement
exceedances. Overall, CTM-predicted a priori information resulted in LMT $X_r$ values with the least bias and bias
standard deviation (-4.8 $\pm$ 4.8 ppb), RMSE (6.71 ppb), and accuracy/precision exceedances.

At the location of the JPL TMF lidar, much larger daily variability in LMT $O_3$ mixing ratios were observed

during the summer of 2014 compared to the other TOLNet systems. LMT $X_r$ values, using all sources of data as a
priori information, had difficulty in replicating this large variability (see Fig. 6). From Table 4, it can be seen that
despite relatively low biases when using all sources of a priori (< 5.0 ppb), the inability of LMT $X_r$ values to capture
the dynamic daily variability resulted in large bias standard deviations and RMSE values (> 12.5 ppb). Furthermore,
6-10 accuracy/precision requirement exceedances out of 26 total days were calculated when using all sources of a
priori. Despite 6 error exceedances (the least of all profile products), applying GEOS-Chem predictions as a priori
information resulted in the lowest mean biases (1.0 ppb) and RMSE values (12.54 ppb). Typically, large
underestimations of LMT $X_r$ values occurred when the lidar observed large $O_3$ enhancements near the surface and
significant overestimations of LMT $X_r$ values were calculated when the lidar observed very large $O_3$ lamina (>150
ppb) aloft. This indicates that the shape of the a priori $O_3$ vertical profile used in TEMPO tropospheric $O_3$ retrievals
are very important in order to capture $X_r$ values for both the tropospheric and LMT column and this will be discussed
in Sect. 3.2.3.
Figure 6 and Table 4 demonstrate that in general $X_r$ values in the troposphere and near the surface are more
accurately retrieved when applying NRT model predictions, and in particular CTM values from GEOS-Chem, at all 3
TOLNet system locations. Also, from this figure it can be seen that in general when large daily-averaged LMT $O_3$
mixing ratios are observed (here defined as days with daily-averaged LMT $O_3 > 65$ ppb), which are important for air
quality purposes, LMT $X_r$ values display less bias when applying GEOS-Chem a priori profile information compared
to all other products. For the 11 days in which daily-averaged LMT $O_3$ mixing ratios exceeded 65 ppb, 64%, 9%, and
27% of the LMT $X_r$ values had the smallest bias using GEOS-Chem, GEOS-5 FP, and MERRA2 a priori profiles,
respectively. This suggests that applying NRT CTM predictions as a priori profile information will allow TEMPO to
observe air quality relevant pollution concentrations of LMT $O_3$ more accurately compared to TB-Clim and models
with simplistic/limited atmospheric chemistry schemes and emission inventories evaluated during this work.
**3.2.3 Importance of a priori vertical profile shape**
Figure 7 displays examples of why climatological a priori information in theoretical TEMPO retrievals resulted in
large daily-averaged LMT column $X_r$ biases. The first example in Fig. 7a shows the daily-averaged vertical profiles
of $X_a$ and $X_r$ with the use of TB-Clim and GEOS-Chem a priori on 08 July 2014 at the JPL TMF site when the lidar
observed large LMT $O_3$ values above EPA NAAQS levels. This case study illustrates how CTMs are more likely to
be able to replicate surface $O_3$ enhancements compared to climatological products. The GEOS-Chem a priori
information resulted in more accurate TEMPO $X_r$ values for the tropospheric and LMT $O_3$ column values. When using
GEOS-Chem model predictions as a priori information, TEMPO LMT column $X_r$ retrievals (65.1 ppb) were closer in
magnitude to observations (70.2 ppb) compared to when using TB-Clim a priori (54.7 ppb). Furthermore, when using
GEOS-Chem a priori information, TEMPO retrievals for the troposphere (65.8 ppb) were also more similar in
magnitude to lidar observations (64.2 ppb) compared to using a priori data from TB-Clim (68.2 ppb).
Another example is illustrated in Fig. 7b which shows $X_a$ and $X_r$ when using TB-Clim and GEOS-5 FP
predictions as a priori profiles in TEMPO retrievals on 21 August 2014 at the JPL TMF lidar location. On this day, a
STE event was likely occurring as tropospheric $O_3$ mixing ratios were measured to be $> 200$ ppb between 6-9 km.
This case study illustrates how a NRT DAS model, GEOS-5 FP, displayed some ability to replicate the large $O_3$ lamina
in the middle/upper troposphere due to being constrained with upper atmospheric observations. The GEOS-5 FP a
priori information resulted in more accurate TEMPO $X_r$ values for the tropospheric and LMT $O_3$ column values. When
using GEOS-5 FP data as a priori information, TEMPO $X_r$ values for tropospheric $O_3$ of 130.4 ppb compared closely
to the JPL TMF lidar observations (135.6 ppb) while TB-Clim data resulted in much lower values (112.4 ppb).
However, the large adjustment needed to correct the a priori profiles to match tropospheric column $O_3$ observations
led to noticeable overestimations of TEMPO LMT $X_r$ values. Since the GEOS-5 FP a priori data was able to better
replicate the STE event compared to TB-Clim, the LMT $X_r$ overestimation of observed LMT $O_3$ values (48.8 ppb) is
much less when applying GEOS-5 FP (77.6 ppb) than when applying TB-Clim (99.1 ppb).

Overall, these results demonstrate that because TEMPO will only have up to ~1.5 DFS in the troposphere

(only ~0.2-0.4 DFS in the 0-2 km level), it is important for a priori profiles to match the general shape of observations,
throughout the entire troposphere and LMT, in order to accurately retrieve both total tropospheric and LMT $O_3$ values.
While the magnitude of the tropospheric $O_3$ column will be largely controlled by the retrieval, the shape of the a priori
profile itself will have an impact on the shape of the retrieved tropospheric $O_3$ profile, and therefore the LMT $O_3$
magnitudes where satellite sensitivity is low.
**3.2.4 Diurnal cycle of tropospheric TEMPO retrievals**
This section focuses on evaluating the ability of TEMPO to retrieve hourly-averaged tropospheric $O_3$ applying TB-
Clim and the GEOS-5 FP, MERRA2, and GEOS-Chem models as a priori profile information. This evaluation was
conducted for one day each at the RO3QET and TROPOZ sites when constant lidar measurements were obtained in
the troposphere/LMT and near-surface $O_3$ enhancements with potential air quality relevant impacts were observed.
Figure 8 shows the time-series of hourly-averaged tropospheric and LMT column $X_r$ retrievals when using TB-Clim
and models as a priori compared to that observed by RO3QET on 07 August 2014 and by TROPOZ on 22 July 2014.
This figure also displays the a priori vertical $O_3$ profiles used in TEMPO retrievals for the hour of largest LMT $O_3$
observations from the TOLNet systems (20 UTC at the RO3QET location and 22 UTC at the TROPOZ site location).

In comparison to lidar measurements by RO3QET, TEMPO retrievals, with all sources of a priori profiles,

are able to reproduce the diurnal pattern of tropospheric and LMT column $O_3$ values (all R values > 0.98) (see Table
5 and Fig. 8). Table 5 shows that all a priori products resulted in TEMPO retrieving average tropospheric column $O_3$
with minimal biases, however, GEOS-Chem was the only product which resulted in LMT $X_r$ values comparable to
observations. This is because GEOS-Chem a priori profiles allow for more dynamic $O_3$ retrievals for the entire
troposphere and LMT. This is demonstrated by the fact that the daily-mean and standard deviation (1σ) of hourly LMT
$O_3$ from TEMPO using GEOS-Chem a priori information (62.1 ± 5.4 ppb) compared the closest to RO3QET
observations (65.2 ± 9.3 ppb). The daily-mean and standard deviations for LMT $X_r$ retrievals, using the other a priori
profiles, underpredicted the magnitude and diurnal variability to a higher degree compared to predictions using GEOS-
Chem a priori.

Similar results are displayed in Fig. 8 and Table 5 when evaluating the case study at the TROPOZ site

location. Once again, TEMPO retrievals with all sources of a priori profiles are generally able to reproduce the diurnal
pattern of tropospheric and LMT column $O_3$ values (all R values ≥ 0.51) but all show large negative biases compared
to LMT observations. These low biases are likely due to the very large LMT $O_3$ values measured by TROPOZ on this
day associated with complex vertical/horizontal transport (Sullivan et al., 2016) which were not well reproduced by a
priori products evaluated during this study. However, Table 5 shows that the GEOS-Chem model a priori data resulted
in TEMPO retrievals of hourly tropospheric and LMT $O_3$ with the least bias. LMT $X_r$ values using the TB-Clim,
GEOS-5 FP, and MERRA2 a priori information displayed too little diurnal variability (nearly a factor of 2 lower
standard deviation compared to TEMPO retrievals using GEOS-Chem a priori data) and a consistent underestimate
of observations. During both case studies, a priori profile shape was critical for TEMPO retrievals to accurately
retrieve both tropospheric and LMT $O_3$. Figure 8 shows a priori profiles from all products for the hour of each day
when largest LMT $O_3$ observations occurred. This figure further emphasizes that GEOS-Chem CTM simulations are
able to better capture the dynamic vertical $O_3$ profiles observed by the lidars compared to the other a priori profile
sources. While the GEOS-Chem $X_a$ profiles underestimate the large LMT $O_3$ enhancements, the ability to replicate
the general shape greatly improves tropospheric and LMT column TEMPO $X_r$ values.
**4 Conclusions**
This study evaluated the a priori vertical $O_3$ profile product currently suggested to be used in TEMPO tropospheric
profile retrievals (TB-Clim, Zoogman et al., 2017) and simulated profiles from operational (GEOS-5 FP), reanalysis
(MERRA2), and CTM predictions (GEOS-Chem). The spatio-temporal representativeness of the vertical profiles from
each product was evaluated using TOLNet lidar observations of tropospheric $O_3$ during the summer (July-August) of
2014. The TOLNet sites used in this study were situated in areas which represent the southeastern US (RO3QET),
intermountain west (TROPOZ), and remote high-elevation locations in the western US (JPL TMF). Because TEMPO
will provide high spatial resolution tropospheric (0-10 km) and LMT (0-2 km) $O_3$ values on an hourly time scale,
potential sources of a priori profiles must be able to replicate inter-daily variability and the diurnal cycle of observed
vertical tropospheric $O_3$ profiles.
When evaluating summertime-averaged tropospheric $O_3$ profiles, it was found that TB-Clim, GEOS-5 FP,
MERRA2, and GEOS-Chem data could generally replicate the vertical structure of tropospheric $O_3$ measured by
TOLNet lidars. However, the seasonal/monthly evaluation is insufficient as TEMPO will provide hourly, high spatial
resolution, tropospheric and LMT $O_3$ values. The evaluation of daily-averaged tropospheric and LMT column $O_3$
values from these products using lidar observations resulted in varying statistical comparisons. Overall, at all 3
TOLNet system locations, GEOS-Chem provided the only data product which consistently captured the inter-daily
variability of tropospheric and LMT column $O_3$ observations. Furthermore, due to the monthly- and zonal-mean nature
of TB-Clim, this product was unable to reproduce the inter-daily variability of tropospheric $O_3$. The ability of the NRT
models, in particular GEOS-Chem, to better replicate the temporal variability of $O_3$ observations led to better statistical
comparisons to daily-averaged TOLNet data. An important fact demonstrated in this study is that models, primarily
GEOS-Chem CTM predictions, displayed better skill in reproducing the largest peaks in daily-averaged near surface
$O_3$ observations which have important implications for air quality. This is partially because GEOS-Chem data best
replicated the diurnal cycle of observations of tropospheric and LMT column $O_3$. Overall, the GEOS-Chem CTM
predictions had the best statistical comparison to daily- and hourly-averaged tropospheric and LMT column $O_3$
observations.
The impact of different a priori profile products on TEMPO tropospheric $O_3$ retrievals was evaluated during
this study. The results demonstrate that since TEMPO will only have up to ~1.5 DFS in the troposphere (and ~0.2-0.4
in the 0-2 km column), the ability of the a priori profile to replicate the general shape of the "true" $O_3$ vertical structure
(throughout the entire troposphere and LMT) is important in order for the sensor to accurately retrieve both
tropospheric column and near surface $O_3$ values. In general, the magnitude of the tropospheric $O_3$ column from
TEMPO will be largely controlled by the retrieval and the shape of the a priori profile will have a noticeable impact
on the shape of the retrieved tropospheric $O_3$ profile, and therefore the LMT $O_3$ magnitudes where satellite sensitivity
is low. This was demonstrated as TEMPO $X_r$ values, using all a priori data, were able to accurately retrieve highly
variable column tropospheric $O_3$ magnitudes, however, large differences in LMT $X_r$ values were calculated. In
general, LMT column $X_r$ values were more accurately retrieved with model a priori profiles, especially with GEOS-
Chem predictions. The better performance of TEMPO LMT $X_r$ values, with GEOS-Chem a priori profiles, is because
it better reproduces the dynamic vertical structures and inter-daily/diurnal variability of tropospheric $O_3$. Most
importantly from an air quality perspective is that when large daily-averaged LMT $O_3$ mixing ratios were observed,
$X_r$ values near the surface with GEOS-Chem a priori displayed the least bias. Overall, this study suggests that applying
a NRT CTM as a priori will likely allow TEMPO retrievals to observe air quality relevant $O_3$ concentrations more
accurately than TB-Clim and other models with limited atmospheric chemistry schemes and emission inventories.
This study is a first step in determining the impact of varying a priori profile sources on the accuracy of
TEMPO tropospheric and LMT column $O_3$ retrievals in North America. The results demonstrate that model
simulations, in particular those from a CTM, improve TEMPO tropospheric $O_3$ retrievals over climatological products
such as TB-Clim when applied as the a priori. However, there are instances where CTM predictions did not improve
TEMPO retrieved values compared to the TB-Clim data. Furthermore, out of the 59 total days of TOLNet observations
analyzed during this study, LMT column $X_r$ values using GEOS-Chem a priori profiles show biases greater than the
TEMPO 10 ppb accuracy requirement for ~15% of the days. It should be noted that this number of LMT column $X_r$
error exceedances is the least compared to when using all the sources of a priori and greater than a factor of 2 smaller
than when applying TB-Clim a priori. The main reason for the majority of error exceedances is because the a priori
profiles do not capture the dynamic vertical $O_3$ profile observed by the TOLNet lidars.
The results of this study clearly demonstrate that using simulated NRT (non-climatological) $O_3$ profile data
will improve near-surface TEMPO $O_3$ retrievals, however, implementing NRT daily/hourly predictions from CTM or
air quality models as the a prior is best suited for using TEMPO data to study topics such as air quality or event-based
processes (e.g., air quality exceedances, wildfires, stratospheric intrusions, pollution transport, etc.). Applying NRT
daily/hourly predictions from CTM or air quality models as the a priori will impact errors/uncertainties and long-term
trends in tropospheric $O_3$ retrievals from TEMPO and these impacts would be difficult to separate from actually
retrieved information. Therefore, the standard TEMPO $O_3$ profile algorithm will need to use an hourly-resolved
monthly mean climatology and follow-on studies to this manuscript are currently being conducted to develop different
CTM-simulated $O_3$ climatology products and test them in the retrieval algorithm. It is important to note that TEMPO
data users can easily apply the output from the standard retrieval (e.g., original a priori $O_3$ profile, retrieved $O_3$ profile,
and AKs) and recalculate the tropospheric $O_3$ vertical profiles using a new/different source of a priori following the
methods of this study. This will allow data users to apply a priori profiles they believe will result in the most
accurate/representative tropospheric and LMT $O_3$ magnitudes from TEMPO without having to rerun the
computationally-expensive SAO retrieval algorithm.
**Appendix A: Testing the linearity assumption applied in Eq. (1)**
The linear estimation technique applied in this study, presented in Eq. (1), utilizes pre-computed AKs (described in
Sect. 2.2.1). This appendix is designed to test whether applying a priori profiles, which differ from the original $O_3$
profile applied to calculate these pre-computed AKs, results in: 1) a breakdown of the near-linear assumption
necessary for Eq. (1) or 2) estimates of retrieved $O_3$ profiles which differ drastically from those calculated in the full
non-linear iterative TEMPO retrieval algorithm. To achieve this, we produced pre-computed AKs, using the TEMPO
retrieval algorithm and a TB-Clim a priori profile (referred to as the "normal" a priori throughout Appendix A) and
the error covariance matrix, and then applied these AKs with "extreme" a priori $O_3$ profiles in the linear estimation
technique and compared the output to results from the full non-linear TEMPO retrieval (using the same "extreme" a
priori $O_3$ profiles). These sensitivity tests are representative of the methods applied in this study and will test whether
they had any noticeable impact on the results of this work. Figure A1a shows the a priori profiles which were applied
in the sensitivity study and the resulting DFS calculated in the full non-linear TEMPO retrieval algorithm. This figure
shows the normal and 4 extreme a priori profiles which were calculated using altitude-dependent scaling factors
(varying from 2.0/1.5/0.5/0.6 at 16.25 km to 0.5/0.6/2.0/1.5 at 0.25 km) applied to the normal profile. The profiles
were calculated in this way to synthetically produce profiles which differ from the normal a priori by up to 100% in
different altitude ranges. In order to quantify the AKs dependence on the a priori $O_3$ profile choice, we compared the
AKs in terms of tropospheric (0-10 km) and LMT (0-2 km) DFSs calculated from the full non-linear TEMPO retrieval
algorithm when applying the varying a priori profiles. As shown in Fig. A1a, using a priori $O_3$ profiles that differed
by up to 100%, compared to the normal a priori, led to minimal differences (< 1%) in the tropospheric and LMT DFS
values. This is because AKs are not directly related to the a priori profiles but based on the weighting functions derived
from the final retrieval that is only initialized with the a priori profile. Overall, this demonstrates that it is valid to
assume that pre-computed AKs, such as those used in this study, can be applied linearly with different sources of a
priori profiles.

To further investigate the linearity assumption applied when using Eq. (1), we compare $O_3$ values produced
in the non-linear TEMPO retrieval algorithm and those from our linear estimation technique, using pre-computed AKs
calculated applying the normal a priori, when using the normal/extreme a priori profiles shown in Fig. A1a (resulting
retrievals shown in Fig. A1b). From Fig. A1b it can be seen that the linear estimation technique, using pre-computed
AKs, is a good representation of the full non-linear TEMPO retrieval. This is demonstrated by tropospheric and LMT
column total $O_3$ values on average differing by < 10% when using the linear estimation technique and the full non-
linear TEMPO retrieval algorithm. Figure A2 presents the histogram of the percent differences between the $O_3$
calculated from the linear estimation technique and TEMPO retrieval algorithm for all the sensitivity studies at all
vertical levels. This figure shows that the differences between the $O_3$ calculated using Eq. (1), and pre-computed AKs,
and the non-linear TEMPO retrieval algorithm are normally distributed with a peak centered around 0% (mean bias =
0.93%). Furthermore, 74% of co-located comparisons fell within 1σ (5.68%) of the mean percent difference. These
results demonstrate that using highly varying a priori profiles, with the pre-computed AKs used in this study, will
result in $O_3$ values similar to those from the full non-linear retrieval algorithm, thus justifying the near-linear
assumption applied to Eq. (1). It should also be noted that the statistical analysis presented here represents an upper
bound of potential bias as only a priori profiles with very large differences compared to the normal profile are applied.
Therefore, for our study, average biases can be assumed to be < 5-10% when the a priori profile differs greatly (around
a factor of 2) from an average $O_3$ vertical profile. Overall, the results of the sensitivity studies presented here suggest
that the linearity assumption applied to Eq. (1) in this study is valid and will have minimal impact on the results of
this study as: 1) extremely small differences in sensitivity (AKs) are computed with highly varying a priori profiles
compared to the "normal" a priori data, 2) rarely do a priori profiles used in this study differ by up to 100% (Fig. 7
illustrates the cases where a priori profiles differ the largest in this study), and 3) when a priori profiles differ by nearly
a factor of 2, the resulting retrieved profiles using Eq. (1) differ by much larger values than the small potential biases
presented here (see Fig. 7b).
*Acknowledgements*. This work is supported by the TOLNet program within NASA's Science Mission Directorate. X.
Liu and P. Zoogman were supported by the NASA Earth Venture Instrument TEMPO project (NNL13AA09C). The
authors would also like to thank the Harvard University Atmospheric Chemistry Modeling Group for providing the
GEOS-Chem model and the NASA GMAO for providing the GEOS-5 FP and MERRA2 products used during our
research. Resources supporting this work were provided by the NASA High-End Computing (HEC) Program through
the NASA Advanced Supercomputing (NAS) Division at NASA Ames Research Center. All the authors express
gratitude to the support from NASA's Earth Science Division at Ames Research Center. Finally, the views, opinions,
and findings contained in this report are those of the authors and should not be construed as an official NASA or
United States Government position, policy, or decision.

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

**Tables**

**Table 1. Information about the TOLNet systems applied during this study.**

| System Name | Latitude (°N) | Longitude (°W) | Elevation (m)[a] | # of observations[b] |
|---|---|---|---|---|
| TROPOZ | 40.6 | 105.1 | 1569.0 | 21 |
| JPL TMF | 34.4 | 117.7 | 2285.0 | 26[c] |
| RO3QET | 34.7 | 86.6 | 206.0 | 12[d] |

[a]Elevation of the topography above sea level.
[b]Number of days of lidar observations between July - August 2014.
[c]JPL TMF lidar observations only taken during nighttime hours between July-August 2014.
[d]RO3QET lidar observations only taken from the surface to ~5 km agl between July-August 2014.

**Table 2. Time-series evaluation of TB-Clim, GEOS-5 FP, MERRA2, and GEOS-Chem daily-averaged tropospheric and LMT column O₃ with the RO3QET, TROPOZ and JPL TMF lidars. The statistics include correlation (R), mean bias, bias standard deviation (1σ), and root mean squared error (RMSE).**

| RO3QET | TB-Clim | GEOS-5 FP | MERRA2 | GEOS-Chem |
|---|---|---|---|---|
| *Tropospheric Column O₃ (0-5 km)* | | | | |
| Correlation (R) | -0.09 | 0.23 | -0.10 | 0.61 |
| Bias ± 1σ (ppb) | 3.7 ± 6.0 | 2.8 ± 5.6 | -0.7 ± 5.8 | 1.7 ± 4.2 |
| RMSE (ppb) | 6.81 | 6.14 | 5.61 | 4.34 |
| *LMT Column O₃ (0-2 km)* | | | | |
| Correlation (R) | -0.68 | 0.03 | -0.19 | 0.83 |
| Bias ± 1σ (ppb) | 2.9 ± 9.7 | -2.9 ± 8.5 | -4.9 ± 8.0 | -1.3 ± 4.4 |
| RMSE (ppb) | 9.75 | 8.65 | 9.06 | 4.39 |
| **TROPOZ** | **TB-Clim** | **GEOS-5 FP** | **MERRA2** | **GEOS-Chem** |
| *Tropospheric Column O₃ (0-10 km)* | | | | |
| Correlation (R) | -0.09 | 0.26 | 0.38 | 0.82 |
| Bias ± 1σ (ppb) | 2.2 ± 9.7 | 3.3 ± 10.0 | -4.6 ± 9.1 | 2.4 ± 6.0 |
| RMSE (ppb) | 9.73 | 10.33 | 9.99 | 6.30 |
| *LMT Column O₃ (0-2 km)* | | | | |
| Correlation (R) | -0.15 | -0.09 | -0.18 | 0.47 |
| Bias ± 1σ (ppb) | -11.1 ± 7.5 | -4.4 ± 7.3 | -7.4 ± 7.4 | -6.7 ± 6.2 |
| RMSE (ppb) | 13.23 | 8.43 | 10.33 | 8.93 |
| **JPL TMF** | **TB-Clim** | **GEOS-5 FP** | **MERRA2** | **GEOS-Chem** |
| *Tropospheric Column O₃ (0-10 km)* | | | | |
| Correlation (R) | -0.35 | 0.76 | 0.80 | 0.72 |
| Bias ± 1σ (ppb) | 0.3 ± 18.7 | -5.0 ± 13.8 | -10.6 ± 13.4 | -0.5 ± 14.6 |
| RMSE (ppb) | 18.38 | 14.41 | 16.86 | 14.29 |
| *LMT Column O₃ (0-2 km)* | | | | |
| Correlation (R) | -0.53 | -0.21 | 0.22 | 0.49 |
| Bias ± 1σ (ppb) | 3.3 ± 13.6 | -2.4 ± 12.7 | -4.0 ± 11.7 | 0.9 ± 10.4 |
| RMSE (ppb) | 13.72 | 12.68 | 12.14 | 10.24 |

**Table 3. Time-series evaluation of the TB-Clim, GEOS-5 FP, MERRA2, and GEOS-Chem hourly-averaged**
**tropospheric and LMT column O$_3$ with the RO3QET, TROPOZ and JPL TMF lidars. The statistics include**
**correlation (R), mean bias, bias standard deviation (1σ), and root mean squared error (RMSE).**

| RO3QET | TB-Clim | GEOS-5 FP | MERRA2 | GEOS-Chem |
|---|---|---|---|---|
| *Tropospheric Column O$_3$ (0-5 km)* | | | | |
| Correlation (R) | -0.54 | -0.55 | -0.51 | 0.68 |
| Bias ± 1σ (ppb) | 3.5 ± 1.4 | 2.6 ± 1.6 | -1.2 ± 1.5 | 2.1 ± 1.1 |
| RMSE (ppb) | 3.77 | 2.98 | 1.86 | 2.37 |
| *LMT Column O$_3$ (0-2 km)* | | | | |
| Correlation (R) | 0.20 | 0.55 | -0.43 | 0.76 |
| Bias ± 1σ (ppb) | 1.9 ± 3.9 | -3.3 ± 3.6 | -5.9 ± 4.0 | 0.3 ± 2.6 |
| RMSE (ppb) | 4.20 | 4.73 | 7.04 | 2.45 |
| **TROPOZ** | **TB-Clim** | **GEOS-5 FP** | **MERRA2** | **GEOS-Chem** |
| *Tropospheric Column O$_3$ (0-10 km)* | | | | |
| Correlation (R) | -0.07 | -0.38 | -0.56 | 0.78 |
| Bias ± 1σ (ppb) | 2.6 ± 2.5 | 3.3 ± 2.6 | -5.1 ± 3.2 | 2.2 ± 1.7 |
| RMSE (ppb) | 3.57 | 4.17 | 6.00 | 2.74 |
| *LMT Column O$_3$ (0-2 km)* | | | | |
| Correlation (R) | 0.26 | 0.76 | 0.67 | 0.92 |
| Bias ± 1σ (ppb) | -12.6 ± 6.9 | -7.5 ± 6.6 | -9.6 ± 6.9 | -7.7 ± 4.8 |
| RMSE (ppb) | 14.25 | 9.91 | 11.70 | 9.01 |

814

**Table 4. Time-series evaluation of daily-averaged $X_r$ predictions using the TB-Clim, GEOS-5 FP, MERRA2, and GEOS-Chem data as a priori information in theoretical TEMPO retrievals of tropospheric and LMT column O₃ values with RO3QET, TROPOZ and JPL TMF lidars. The statistics include correlation (R), mean bias, bias standard deviation (1σ), root mean squared error (RMSE), and the number of occurrences where error exceeds 10 ppb.**

| RO3QET | TB-Clim | GEOS-5 FP | MERRA2 | GEOS-Chem |
|---|---|---|---|---|
| *Tropospheric Column O₃ (0-5 km)* | | | | |
| Correlation (R) | 0.98 | 0.90 | 0.95 | 0.96 |
| Bias ± 1σ (ppb) | 1.4 ± 2.3 | 1.3 ± 2.7 | -0.2 ± 2.5 | 1.0 ± 2.0 |
| RMSE (ppb) | 2.66 | 2.91 | 2.43 | 2.17 |
| 10 ppb error exceedance | 0 | 0 | 0 | 0 |
| *LMT Column O₃ (0-2 km)* | | | | |
| Correlation (R) | 0.52 | 0.65 | 0.73 | 0.94 |
| Bias ± 1σ (ppb) | 0.2 ± 6.1 | -3.8 ± 5.5 | -3.4 ± 5.1 | -2.2 ± 2.5 |
| RMSE (ppb) | 5.88 | 6.44 | 5.97 | 3.26 |
| 10 ppb error exceedance | 1 | 3 | 2 | 0 |
| **TROPOZ** | **TB-Clim** | **GEOS-5 FP** | **MERRA2** | **GEOS-Chem** |
| *Tropospheric Column O₃ (0-10 km)* | | | | |
| Correlation (R) | 0.97 | 0.92 | 0.94 | 0.92 |
| Bias ± 1σ (ppb) | -0.9 ± 4.2 | -0.6 ± 4.8 | -2.2 ± 4.4 | -0.5 ± 2.7 |
| RMSE (ppb) | 4.21 | 4.72 | 4.85 | 2.66 |
| 10 ppb error exceedance | 1 | 1 | 2 | 0 |
| *LMT Column O₃ (0-2 km)* | | | | |
| Correlation (R) | 0.38 | 0.41 | 0.42 | 0.65 |
| Bias ± 1σ (ppb) | -11.4 ± 6.2 | -6.4 ± 6.3 | -5.1 ± 5.9 | -4.8 ± 4.8 |
| RMSE (ppb) | 12.95 | 8.85 | 7.67 | 6.71 |
| 10 ppb error exceedance | 10 | 6 | 4 | 3 |
| **JPL TMF** | **TB-Clim** | **GEOS-5 FP** | **MERRA2** | **GEOS-Chem** |
| *Tropospheric Column O₃ (0-10 km)* | | | | |
| Correlation (R) | 0.98 | 0.99 | 0.99 | 0.99 |
| Bias ± 1σ (ppb) | -0.2 ± 4.0 | -0.8 ± 3.1 | -1.7 ± 3.0 | -0.3 ± 3.3 |
| RMSE (ppb) | 3.97 | 3.14 | 3.42 | 3.29 |
| 10 ppb error exceedance | 1 | 1 | 1 | 1 |
| *LMT Column O₃ (0-2 km)* | | | | |
| Correlation (R) | 0.31 | 0.25 | 0.39 | 0.42 |
| Bias ± 1σ (ppb) | 3.1 ± 14.8 | 1.9 ± 13.7 | 4.8 ± 12.6 | 1.0 ± 12.7 |
| RMSE (ppb) | 14.87 | 13.57 | 13.27 | 12.54 |
| 10 ppb error exceedance | 9 | 8 | 10 | 6 |

**Table 5. Time-series evaluation of hourly-averaged TOLNet observations and $X_r$ predictions using the TB-**
**Clim, GEOS-5 FP, MERRA2, and GEOS-Chem data as a priori information in theoretical TEMPO retrievals**
**of tropospheric and LMT column O₃ values at the location of RO3QET (07 August, 2014) and TROPOZ (22**
**July, 2014). The statistics include correlation (R), mean, min/max, and standard deviation (1σ) from**
**observations and theoretical TEMPO retrievals.**

| RO3QET 07 August, 2014 | TOLNet[*] | TB-Clim | GEOS-5 FP | MERRA2 | GEOS-Chem |
|---|---|---|---|---|---|
| *Tropospheric Column O₃ (0-5 km)* | | | | | |
| Correlation (R) | N/A | 0.99 | 0.99 | 0.99 | 0.99 |
| Mean (ppb) | 60.7 | 59.8 | 59.5 | 59.0 | 59.5 |
| Max/Min (ppb) | 67.5/56.4 | 64.7/56.8 | 64.1/56.9 | 63.8/56.1 | 65.1/55.5 |
| Std. Dev. (ppb) | 3.62 | 2.63 | 2.35 | 2.55 | 3.18 |
| *LMT Column O₃ (0-2 km)* | | | | | |
| Correlation (R) | N/A | 0.98 | 0.98 | 0.99 | 0.98 |
| Mean (ppb) | 65.2 | 56.5 | 53.4 | 53.1 | 62.1 |
| Max/Min (ppb) | 79.4/54.3 | 62.6/52.5 | 59.4/49.8 | 59.4/48.8 | 70.6/54.6 |
| Std. Dev. (ppb) | 9.27 | 3.41 | 3.33 | 3.67 | 5.38 |
| **TROPOZ 22 July, 2014** | **TOLNet** | **TB-Clim** | **GEOS-5 FP** | **MERRA2** | **GEOS-Chem** |
| *Tropospheric Column O₃ (0-10 km)* | | | | | |
| Correlation (R) | N/A | 0.98 | 0.97 | 0.96 | 0.97 |
| Mean (ppb) | 50.5 | 52.4 | 52.2 | 50.7 | 50.3 |
| Max/Min (ppb) | 55.8/46.3 | 55.7/49.2 | 55.5/49.0 | 53.3/47.7 | 53.3/47.3 |
| Std. Dev. (ppb) | 3.25 | 2.60 | 2.52 | 2.06 | 2.40 |
| *LMT Column O₃ (0-2 km)* | | | | | |
| Correlation (R) | N/A | 0.85 | 0.51 | 0.79 | 0.98 |
| Mean (ppb) | 75.0 | 44.3 | 49.9 | 51.2 | 56.3 |
| Max/Min (ppb) | 97.0/58.6 | 47.5/41.3 | 54.3/45.6 | 54.9/47.3 | 66.4/47.8 |
| Std. Dev. (ppb) | 12.77 | 2.27 | 2.96 | 2.81 | 5.93 |

[*]Correlation values are computed between the O₃ climatology and models compared to observations (i.e., TOLNet)
and therefore are presented as N/A for TOLNet.

**Figures**

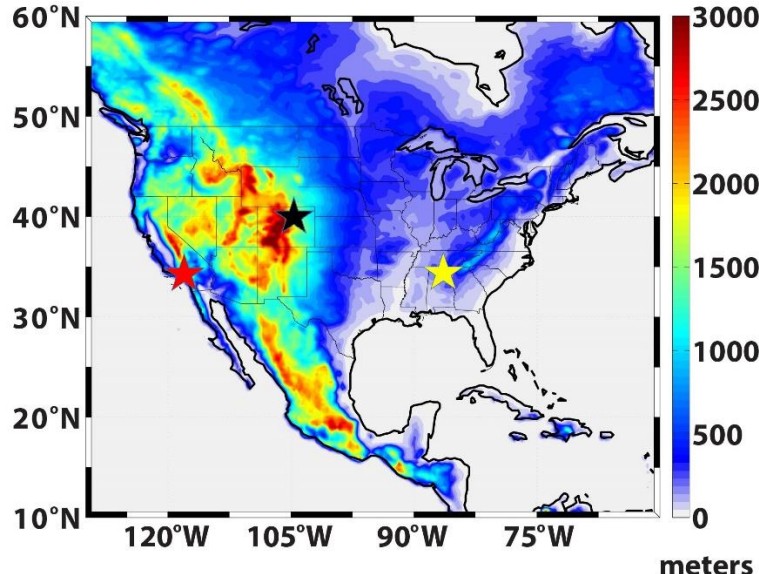


**Figure 1. Location of the GSFC TROPOZ (black star), JPL TMF (red star), and the UAH RO3QET (yellow**
**star) TOLNet systems during the summer of 2014. The locations are overlaid on the topographic heights**
**(meters) from the GEOS-5 model.**

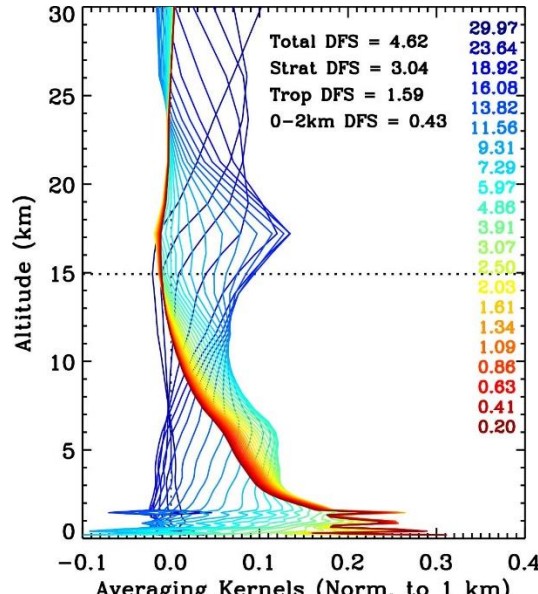

**Figure 2. Simulated TEMPO O₃ retrieval AK matrix (normalized to 1 km layer) from joint UV+VIS**
**measurements (290-345 nm, 540-650 nm) from the surface to 30 km agl used at the UAH TOLNet site during**
**August at 20 UTC. The AK lines are for individual vertical levels (km agl), with the colors ranging from red to**
**blue representing vertical levels from surface air to ~30 km. The legend presents the DFS for the total (Total),**
**stratosphere (Strat), troposphere (Trop), and 0-2 km columns.**

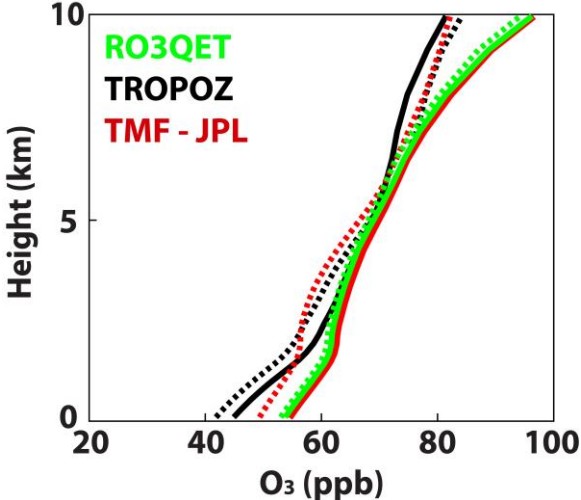


**Figure 3. Monthly-averaged vertical profiles of O$_3$ (ppb) from TB-Clim data at the location of the RO3QET**
**(green lines), TROPOZ (black lines), and JPL TMF (red lines) TOLNet systems for July (solid lines) and August**
**(dashed lines). The monthly-averages are derived using the hourly TB-Clim data during the hours/days of**
**TOLNet observations obtained at each location.**

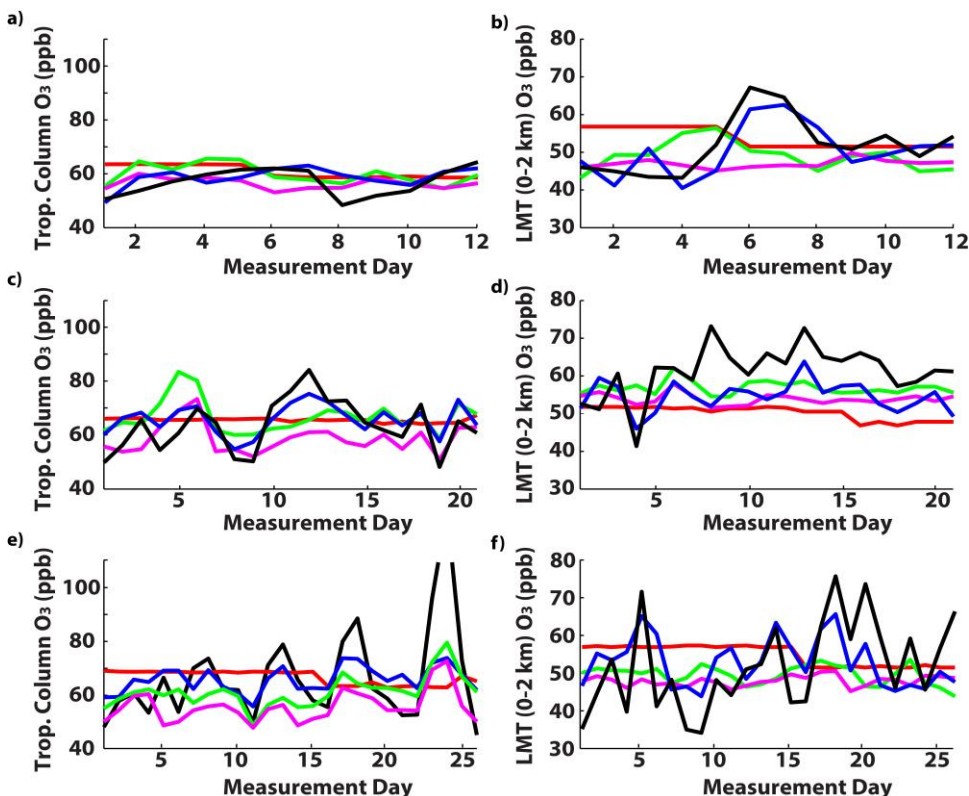


**Figure 4. Time-series of daily-averaged tropospheric column (0-10 km) O$_3$ (ppb) from TB-Clim (red line),**
**GEOS-5 FP (green line), MERRA2 (magenta line), and GEOS-Chem (blue line) compared to TOLNet (black**
**line) at the locations of a) RO3QET, c) TROPOZ, and e) JPL TMF. Panels b), d), and f) are similar but for the**
**comparison of LMT column (0-2 km) O$_3$.**

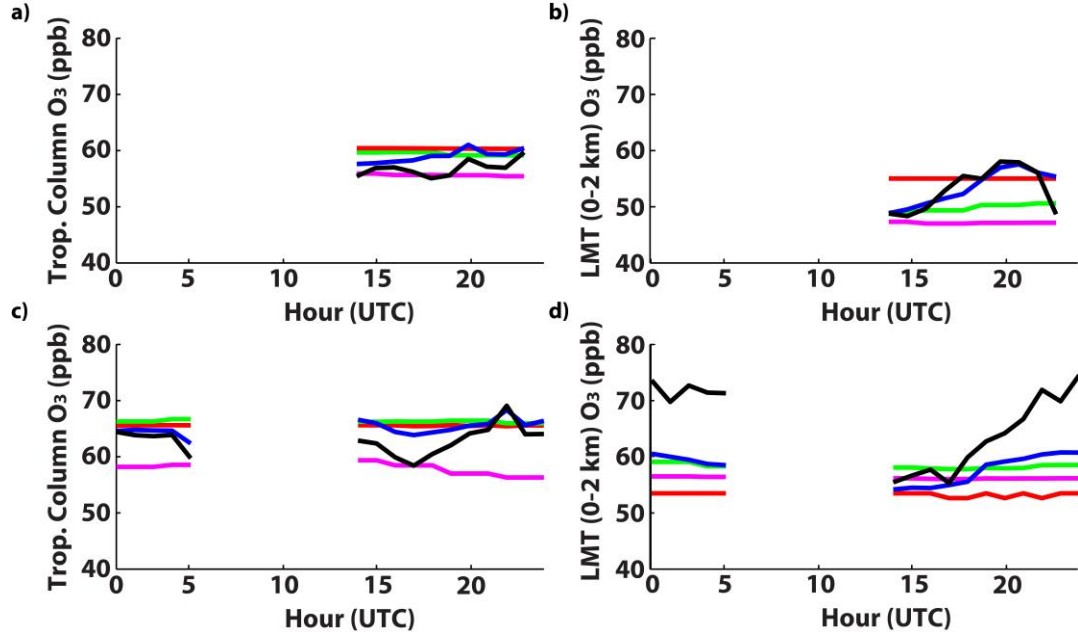


**Figure 5. Diurnal time-series of hourly-averaged tropospheric column (0-10 km) O₃ (ppb) from TB-Clim (red**
**line), GEOS-5 FP (green line), MERRA2 (magenta line), and GEOS-Chem (blue line) compared to TOLNet**
**(black line) at the locations of a) RO3QET and c) TROPOZ. Panels b) and d) are similar but for the comparison**
**of LMT column (0-2 km) O₃. The times of missing data are hours where no TOLNet observations were taken**
**during the summer of 2014.**

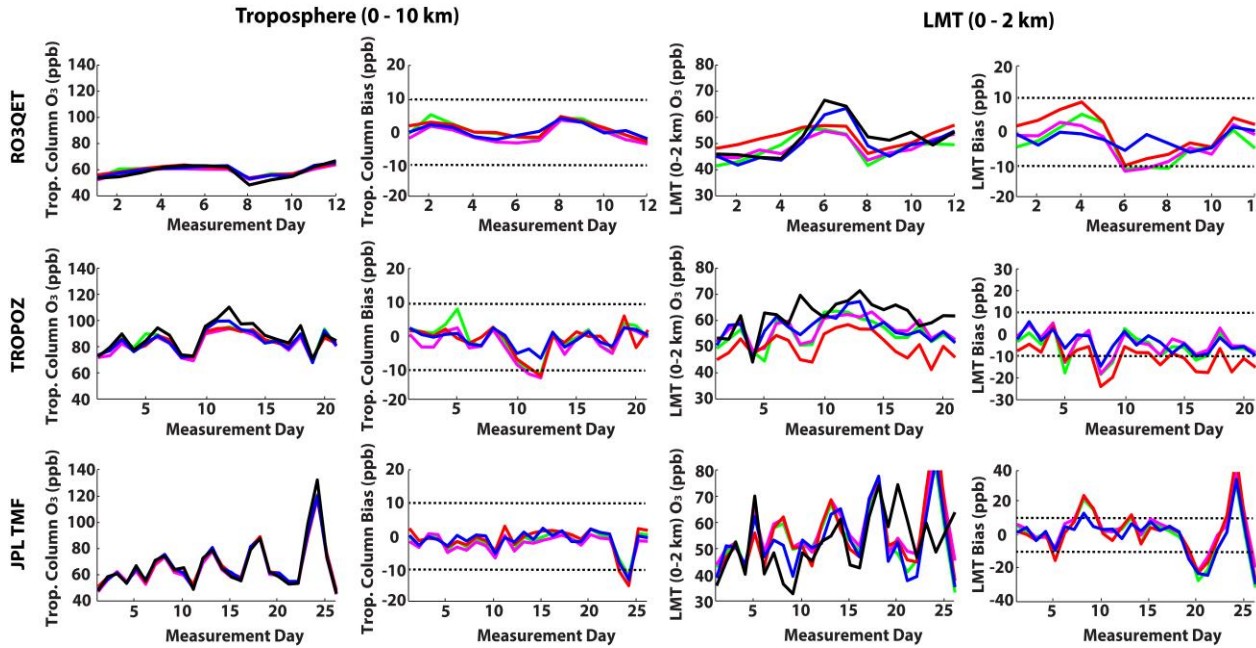


**Figure 6. Time-series of daily-averaged tropospheric and LMT column $X_r$ and bias values (ppb) when using**
**TB-Clim (red line), GEOS-5 FP (green line), MERRA2 (magenta line), and GEOS-Chem (blue line) as the a**
**priori when compared to observed O₃ by TOLNet (black line) at the locations of RO3QET (top row), TROPOZ**
**(middle row), and JPL TMF (bottom row). The dashed black lines represent the 10 ppb precision/accuracy**
**requirement for TEMPO O₃ retrievals.**

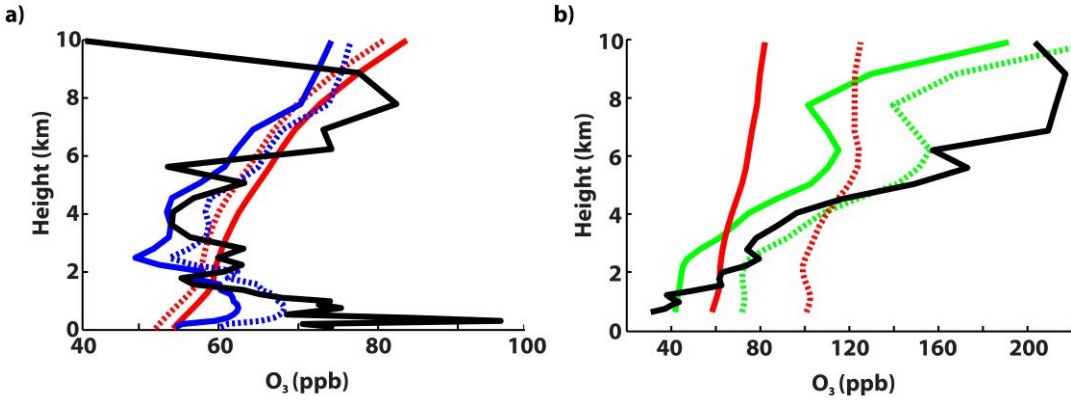


**Figure 7. Vertical profiles of a) daily-averaged $X_a$ (solid line) and $X_r$ (dashed line) O₃ values when applying**
**TB-Clim (red line) and GEOS-Chem (blue line) as a priori information in TEMPO retrievals compared to**
**TOLNet (black line) at the locations of the JPL TMF lidar on 08 July, 2014. Panel b) shows daily-averaged $X_a$**
**and $X_r$ O₃ values when applying TB-Clim (red line) and GEOS-5 FP (green line) as a priori information in**
**TEMPO retrievals compared to TOLNet (black line) at the locations of the JPL TMF lidar on 21 August, 2014.**

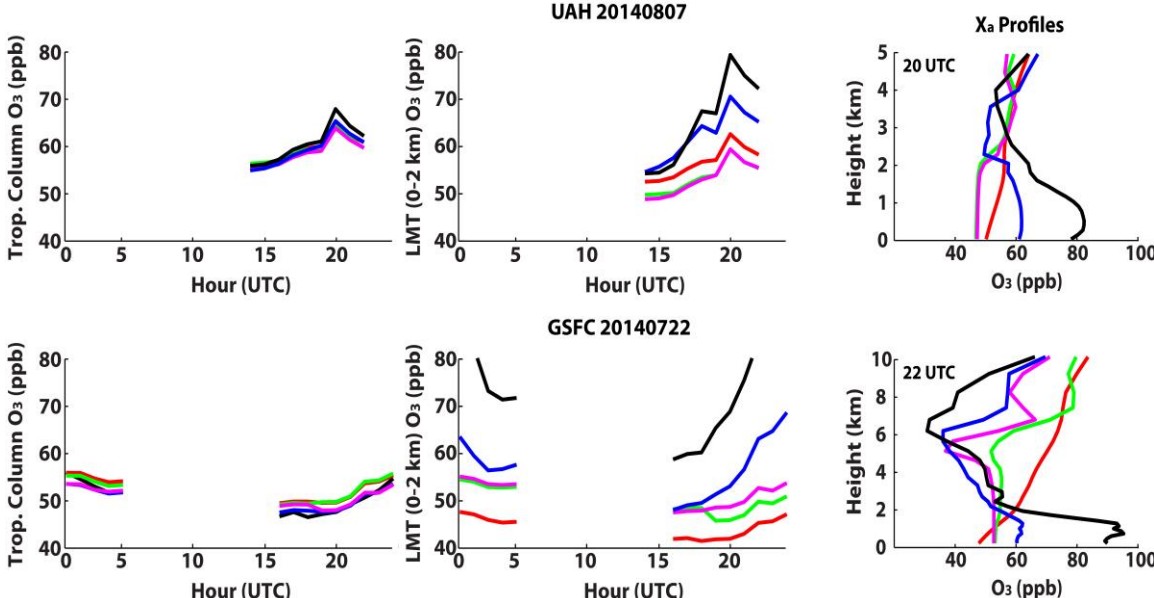


**Figure 8. Diurnal time-series of hourly-averaged tropospheric (0-10 km) and LMT (0-2 km) column $X_r$ O₃**
**(ppb) values with a priori from TB-Clim (red line), GEOS-5 FP (green line), MERRA2 (magenta line), and**
**GEOS-Chem (blue line) compared to TOLNet (black line) at the locations of RO3QET location on 07 August**
**2014 (top row) and TROPOZ on 22 July 2014 (bottom row). The hourly-averaged a priori vertical profiles are**
**also presented (right column) along with TOLNet (black line) for the hour of largest LMT O₃ observed by**
**TOLNet in the time-series.**

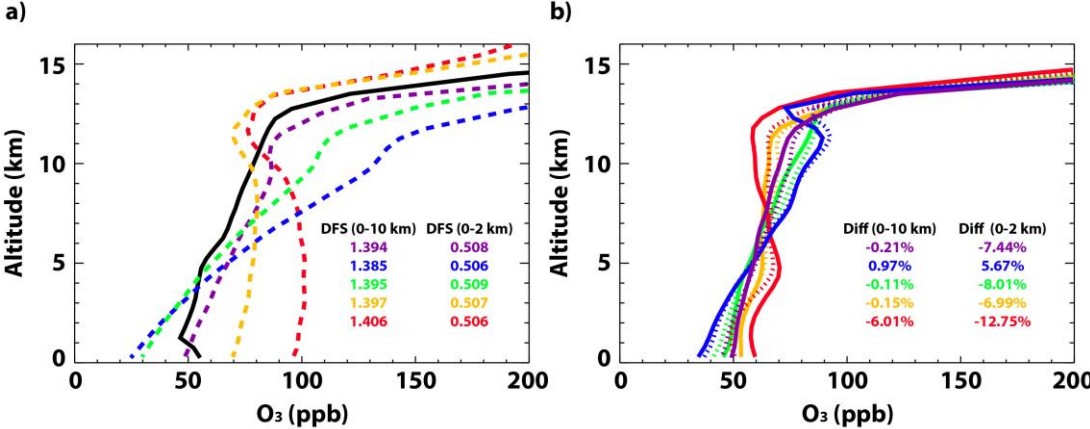


**Figure A1.** Vertical profiles of a) "true" (black solid line), the "normal" TB-Clim a priori (purple dashed line),
and the four "extreme" a priori profiles (other color dashed lines) applied in the sensitivity study to test the
linearity assumption applied in Eq. (1) and b) the retrieved $O_3$ from the full non-linear TEMPO retrieval
algorithm (solid lines) and the linear estimation technique from Eq. (1) (dotted lines). The DFS values were
calculated in the full non-linear TEMPO retrieval algorithm when applying the varying a priori profiles. The
extreme a priori profiles shown here were produced to represent cases were the a priori differs largely from
the TB-Clim data used to produce the pre-computed AKs. These profiles were synthetically produced by
applying altitude-dependent scaling factors (varying from 2.0/1.5/0.5/0.6 at 16.25 km to 0.5/0.6/2.0/1.5 at 0.25
km) to the TB-Clim profile. The color of the lines presented in panel b) indicate which a priori profile was used
in the retrievals. The insets in the figure provide a) the DFS in the troposphere (0-10 km) and LMT (0-2 km)
and b) the percent $O_3$ column difference of calculated $O_3$ between the linear estimation technique and the full
non-linear retrieval algorithm in the troposphere and LMT.


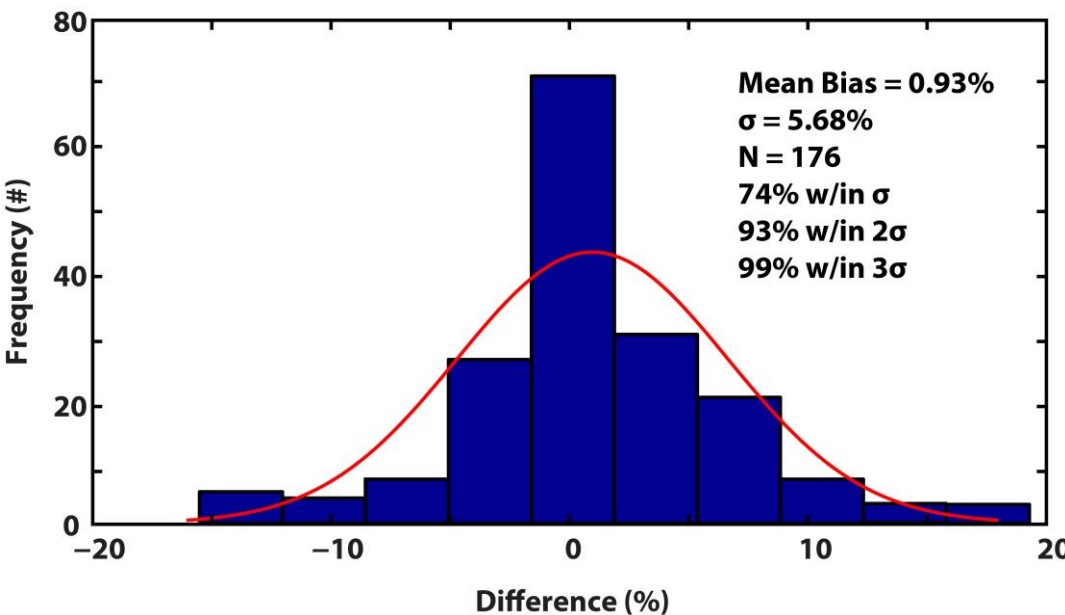


**Figure A2. Statistical comparison of the difference (%) between O₃ calculated using the linear estimation**
**technique and full non-linear TEMPO retrievals. The percent differences are calculated at all vertical levels**
**for the cases using the 4 "extreme" a priori profiles applied during the sensitivity studies. The red line illustrates**
**the normal/gaussian distribution of the percent differences and the inset of the figure provides the statistics of**
**the histogram.**