# Peer review of "Evaluation of potential sources of a priori ozone profiles for TEMPO tropospheric ozone retrievals"

_Atmospheric Measurement Techniques, 2017_

## Referee Comment (RC1) · Anonymous Referee #2 · 6 Feb 2018

In this paper, the authors compare ozone profile data from three TOLNet stations across the USA with A) an ozone climatology and ozone data from various transport models, and B) a simulated retrieval result where the climatology and models are used as a priori values.

The authors use a formula from the book by Rodgers (2000) to linearise the calculations of the effect of the a priori on a potential retrieval. While Rodgers uses the formula in Chap 3 and Chap 10 of his book, using this formula to make a selection on a preferred a priori brushes over the potential issues you often get with real satellite data.

The first question that comes to mind is: how representative are these simulated retrievals for real world situations. Or is Eq 1 limited to be used for an error / sensitivity

study? My impression is that the error component is not used in the paper. Please give your reasons for using this method.

In retrievals of ozone profiles, an a priori consists of a profile shape and an associated error profile. Because the retrieval of ozone profiles is under-determined (more than one profile shape can be retrieved from the same spectrum), an a priori is used in an Optimal Estimation (OE) based retrieval to constrain the outcome to reasonable values. The a priori profile shape is a reference, and the profile error gives the retrieval the freedom to differ from that reference shape based on the input spectrum to minimise the cost function.

In a real retrieval, when either the error on the a priori is set to zero, or the error on the measured / simulated spectrum is too large then the OE retrieval result will reproduce the a priori almost exactly. In this case no information is gained from the spectrum during the retrieval. In other words: the spectrum contains no useful information and the degrees of freedom from the signal (DFS) will be low.

The authors seem to come to the conclusion that TEMPO ozone profile retrievals in the troposphere and LMT require an a priori that already matches the general shape of the observations in order for the required accuracy to be obtained in the retrieval. If the a priori already needs to be so close to the shape and magnitude of the outcome of the retrieval, then one could conclude that the TEMPO spectra do not contain sufficient information for the retrieval, or the retieval is over-constrained.

How do the authors see these issues, in light of the need of their conclusion that the a priori needs to be close to the true profile? Please clarify.

Another way of looking at it is by looking at Eq 1. If the a priori Xa closely matches the true Xt, then what is 'retrieved' is mainly the a priori, as the second term in the equation falls to zero. It is therefore not surprising that an a priori that more closely matches the true profile will also do well in the simulated retrievals. Those a priori profiles already have the advantage in Eq 1. How does this advantage play out with real retrievals? Is

it really necessary to have an a priori so close to the true state of the atmosphere to get a good retrieval? If so, what is the added value of a retrieval in this case?

Textual / other remarks:

Line 104: You mention an error margin of the TOLNet measurements of 10% in the lower troposphere and 20% in the upper troposphere. The words 'lower' and 'upper' are not defined in this context, while you use the terms LMT (0-2km) and tropospheric (0-10km). Please be more specific about the applicable altitude ranges of the errors of the TOLNet DIAL lasers.

In sect 2.2/2.2.1 it would be helpful to have a little more information on the input data. Please elaborate on the setup you use to generate the artificial / simulated TEMPO data (the AK's, the Gain matrices, etc). What other relevant sources of information did you use, like temperature, albedo's, cross sections, solar and viewing angles, reference spectra, etc.

In section 2.2 the authors mention the use / adaptation of the SOA retrieval algorithm for TEMPO to do retrievals. But it is not clear to me whether the SOA algorithm played a role in this paper at all. In the second part of 2.2 a simple vector/matrix based formula is used to calculate the simulated retrieved profile. Did the authors use the SAO model for any of the ozone profile retrievals or was it used in the set-up of the kernels? If it was not used, is it then relevant to for this paper?

Line 141: In Eq 1, there is a component for the effect of noise. Please explain how you treat the last term in the equation. How does this component affect the retrievals and what are the expectations on its effect on the ranking of the a priori sources used?

Line 168 and Fig 3: Yellow is a color that is hard to see on a white background. Please use a color with more contrast.

Line 193: 'due to data constraints'. What kind if data constraints? Is it an issue of lack of sensitivity at the lower troposphere of most existing satellite instruments? Please

clarify.

Line 244: In this section you evaluate the straight model output with the TOLNet profiles, outside the context of use as an a priori. The remark that GEOS-Chem is the 'the only potential source of a priori profiles ...' is out of place here. You address the use of the various models as an a priori in sections 3.2.x.

In lines 248 and 249 the authors give a few aspects that may be the reasons why GEOS-Chem compares better to TOLNet than the other models. It would be insightful to the reader to learn which of these aspects contributes the most to the better comparison.

Section 3.1.2: In this section the authors make an evaluation of how well the climatology and the models can reproduce the daily variability of the lidar measurements. Please elaborate on the time step / time resolution of the models. Is there a reasonable expectation that the models can actually follow the daily cycle, or are the climatology and model fields spaced to far apart in time?

Please consider enlarging your time series plots.
* * *

---

## Referee Comment (RC2) · Anonymous Referee #1 · 26 Feb 2018

The article delivers on its goal of evaluating potential sources of a priori ozone profile information for use in retrievals from TEMPO measurements over North America. The accomplishment is well-summarized by the first sentence of the last paragraph: "This study is a first step in determining what source of a priori vertical O3 profiles should be applied to best enhance the ability of TEMPO to retrieve tropospheric and LMT column O3 in North America."

The retrievals envisioned in the article fall into the best-estimate-for-today category of retrieval approaches. That is, they seek to bring in as much information from climatologies or models or other sources as they can into the final near-real-time product. Such approaches may not be well-suited for climate change studies as it can become difficult to unravel the sources of any trends from the influences of the measurements

versus the influence of the varying a priori profiles. Even with the averaging kernels and a priori profiles provided for each retrieval, assimilation applications of the data will be more complicated too. Do the developers envision that the models will use these retrievals as input to influence the forecasts?

A key performance index for the study is the ability of the retrieved profiles to identify high ozone levels in the lowermost troposphere (LMT 0-2km). With this in mind, Tables 4 and 5 should give correlations so that the readers can better compare the performance of the a priori profiles alone, provided in the earlier tables, to the performance of the retrieved profiles.

I was surprised that the article does not include a discussion of the effects of surface reflectivity (and knowledge of the surface reflectivity and surface pressure) on the lower layer information content. What ground reflectivity was assumed in the clear sky retrievals? How will seasonal variability, especially snow cover, be addressed in the algorithm? A future study could also consider the use of clear versus cloudy or partially cloudy (with cloud height and cloud fraction information from the measurements) results for adjacent pixels to try to identify the below cloud columns better (or even to apply some version of cloud slicing).

Editorial erratum

Table 3 does not contain a listed section for JPL TMF results.
* * *

---

## Author Comment (AC1) · 19 Mar 2018

**Response to Reviewer #1 Comments**

We thank the reviewer for their helpful comments. We have incorporated as many of the reviewers' suggestions as possible into the revised manuscript. All reviewer comments are in italics and the author's responses are in standard font.

The article delivers on its goal of evaluating potential sources of a priori ozone profile information for use in retrievals from TEMPO measurements over North America. The accomplishment is wellsummarized by the first sentence of the last paragraph: "This study is a first step in determining what source of a priori vertical O3 profiles should be applied to best enhance the ability of TEMPO to retrieve tropospheric and LMT column O3 in North America."

The retrievals envisioned in the article fall into the best-estimate-for-today category of retrieval approaches. That is, they seek to bring in as much information from climatologies or models or other sources as they can into the final near-real-time product. Such approaches may not be well-suited for climate change studies as it can become difficult to unravel the sources of any trends from the influences of the measurements versus the influence of the varying a priori profiles. Even with the averaging kernels and a priori profiles provided for each retrieval, assimilation applications of the data will be more complicated too. Do the developers envision that the models will use these retrievals as input to influence the forecasts?

The reviewer brings up an important point. The tropospheric ozone (O3) retrieval algorithm for TEMPO is still under development and testing, and therefore the purpose of this study is to determine the general impact of different sources (climatology products and near-real-time (NRT) data assimilation, reanalysis, and chemical transport model (CTM) data) of a priori profiles on TEMPO tropospheric and lowermost tropospheric (LMT)  $O_3$  retrievals. As the reviewer identifies, implementing NRT daily/hourly predictions from CTM or air quality models as the a prior in tropospheric O3 retrievals from TEMPO is best suited when using this output to study topics such as air quality or event-based processes (e.g., air quality exceedances, wildfires, stratospheric intrusions, pollution transport, etc.). Using an a priori from model-predicted NRT daily/hourly information will in fact impact the error/uncertainties and trends of retrieved tropospheric O3 from TEMPO and the final algorithm will likely use an hourly-resolved monthly mean climatology based on model outputs. Follow-on studies to this manuscript are currently being conducted to develop different CTM-simulated O3 climatology products and test them in the tropospheric O3 retrieval algorithm. To better emphasize these important points, additional text has been added to the updated manuscript, primarily in the conclusions section: "The results of this study demonstrate that using simulated O3 profile data will improve near-surface TEMPO O3 retrievals, however, implementing NRT daily/hourly predictions from CTM or air quality models as the a prior is best suited for using TEMPO data to study topics such as air quality or event-based processes (e.g., air quality exceedances, wildfires, stratospheric intrusions, pollution transport, etc.). Applying NRT daily/hourly predictions from CTM or air quality models as the a priori will impact errors/uncertainties and long-term trends in tropospheric O3 retrievals from TEMPO and these impacts would be difficult to separate from actually retrieved information. Therefore, the

standard TEMPO  $O_3$  profile algorithm will likely use an hourly-resolved monthly mean climatology and follow-on studies to this manuscript are currently being conducted to develop different CTM-simulated  $O_3$  climatology products and test them in the retrieval algorithm. It is important to note that TEMPO data users can easily use the output from the standard retrieval and recalculate the tropospheric  $O_3$  vertical profiles using a different source of a priori following the methods of this study.".

A major application of TEMPO products is envisioned to be the assimilation of the  $O_3$  data (and other chemical constituents) into CTM and air quality models to improve retrospective analysis and forecasts of air quality and tropospheric chemical composition. The standard product of TEMPO  $O_3$  retrievals, or recalculated profiles using a different a priori following the methods of this study, can be assimilated into CTM or air quality models.

A key performance index for the study is the ability of the retrieved profiles to identify high ozone levels in the lowermost troposphere (LMT 0-2km). With this in mind, Tables 4 and 5 should give correlations so that the readers can better compare the performance of the a priori profiles alone, provided in the earlier tables, to the performance of the retrieved profiles.

We agree with the reviewer and these correlation values have been added for tropospheric and LMT column  $O_3$ . Additional text has also been added to the updated manuscript to explain these results.

I was surprised that the article does not include a discussion of the effects of surface reflectivity (and knowledge of the surface reflectivity and surface pressure) on the lower layer information content. What ground reflectivity was assumed in the clear sky retrievals? How will seasonal variability, especially snow cover, be addressed in the algorithm? A future study could also consider the use of clear versus cloudy or partially cloudy (with cloud height and cloud fraction information from the measurements) results for adjacent pixels to try to identify the below cloud columns better (or even to apply some version of cloud slicing).

We agree with the reviewer that near-surface O3 retrievals from ultraviolet + visible (UV+VIS) wavelengths are sensitive to surface reflectance/albedo (primarily in the VIS). TEMPO retrieval sensitivity studies which produced the averaging kernels (AK) used during this study (see Zoogman et al. (2017)) applied surface albedo values from the Global Ozone Monitoring Experiment (GOME) albedo database and surface pressure was taken from the GEOS-5 meteorological model. The GOME database provides a monthly mean surface albedo climatology at a spatial resolution of  $1^{\circ} \times 1^{\circ}$  for multiple wavelengths (from 335 to 772 nm) which were interpolated/extrapolated to match TEMPO retrieved wavelengths. The spatio-temporal variability of snow-cover is taken into account when producing TEMPO AKs, but for this study, which is focused on summer-months, will not have any impact on the results. Some text has been added to updated manuscript to reflect this information: "Surface albedo is taken into account using the GOME albedo database interpolated to match TEMPO wavelengths." and "For detailed

information about the TEMPO retrieval sensitivity studies, and the input variables, used to derive AKs applied during this study see Zoogman et al. (2017).". In the actual TEMPO retrieval, surface albedo will be retrieved as a first-order polynomial in the UV following Liu et al. (2005, 2010) and a new climatology of visible surface albedo spectra has been developed for fitting surface albedo spectra in the visible using multiple parameters (Zoogman et al., 2016). Surface albedo is typically well retrieved from this algorithm and its effect on the retrieval sensitivity/information content is taken into account.

We also agree with the reviewer that a future study focused on the impact of clouds (e.g., fraction height, etc.) would be interesting.

**Editorial erratum**

**Table 3 does not contain a listed section for JPL TMF results.**

Table 3 does not have a listed section for JPL TMF results as this table presents the statistics of the comparison of the diurnal time-series of hourly-averaged tropospheric and LMT O3 from the climatology and models to observations. No hourly-averaged lidar observations were available from the JPL TMF system for diurnal time-series evaluation as stated in Sect. 2.1 of the manuscript: "During the summer of 2014, the JPL TMF lidar only conducted measurements during the nighttime hours and therefore will only be used for daily-averaged comparisons to TB-Clim and model predictions". To better explain this, the updated manuscript in Sect. 2.4 now reads: "Due to the hours of operation, the evaluation at the JPL TMF lidar location was not conducted for hourly-averages and is only applied for summer- and daily-averages."

**References**

- Liu, X., Chance, K., Sioris, C. E., Spurr, R. J. D., Kurosu, T. P., Martin, R. V., and Newchurch, M. J.: Ozone profile and tropospheric ozone retrievals from the Global Ozone Monitoring Experiment: Algorithm description and validation, J. Geophys. Res.-Atmos., 110, D20307, doi:10.1029/2005JD006240, 2005.
- Liu, X., Bhartia, P. K., Chance, K., Spurr, R. J. D., and Kurosu, T. P.: Ozone profile retrievals from the Ozone Monitoring Instrument, Atmos. Chem. Phys., 10, 2521-2537, doi:10.5194/acp-10-2521-2010, 2010.
- Zoogman, P., X. Liu, K. Chance, Q. Sun, C. Schaaf, T. Mahr, T. Wagner, A climatology of visible surface reflectance spectra, submitted to J. Quant. Spectro. & Radiat. Transfer, 180, 39-46, doi:10.1016/j.jqsrt.2016.04.003, 2016.
- Zoogman, P., Liu, X., Suleiman, R., Pennington, W., Flittner, D., Al-Saadi, J., Hilton, B., Nicks, D., Newchurch, M., Carr, J., Janz, S., Andraschko, M., Arola, A., Baker, B., Canova, B., Miller, C. C., Cohen, R., Davis, J., Dussault, M., Edwards, D., Fishman, J., Ghulam, A., Abad,

G. G., Grutter, M., Herman, J., Houck, J., Jacob, D., Joiner, J., Kerridge, B., Kim, J., Krotkov, N., Lamsal, L., Li, C., Lindfors, A., Martin, R., McElroy, C., McLinden, C., Natraj, V., Neil, D., Nowlan, C., O'Sullivan, E., Palmer, P., Pierce, R., Pippin, M., Saiz-Lopez, A., Spurr, R., Szykman, J., Torres, O., Veefkind, J., Veihelmann, B., Wang, H., Wang, J., and Chance, K.: Tropospheric emissions: Monitoring of pollution (TEMPO), J. Quant. Spectrosc. Ra., 186, 17-39, https://doi.org/10.1016/j.jqsrt.2016.05.008, 2017.

---

## Author Comment (AC2) · 19 Mar 2018

**Response to Reviewer #2 Comments**

We thank the reviewer for their helpful comments. We have incorporated as many of the reviewers' suggestions as possible into the revised manuscript. All reviewer comments are in italics and the author's responses are in standard font.

*In this paper, the authors compare ozone profile data from three TOLNet stations across the USA with A) an ozone climatology and ozone data from various transport models, and B) a simulated retrieval result where the climatology and models are used as a priori values.*

*The authors use a formula from the book by Rodgers (2000) to linearise the calculations of the effect of the a priori on a potential retrieval. While Rodgers uses the formula in Chap 3 and Chap 10 of his book, using this formula to make a selection on a preferred a priori brushes over the potential issues you often get with real satellite data.*

*The first question that comes to mind is: how representative are these simulated retrievals for real world situations. Or is Eq 1 limited to be used for an error / sensitivity study? My impression is that the error component is not used in the paper. Please give your reasons for using this method.*

The reviewer is correct in fact that the application of Eq. (1) in our study is representative of a sensitivity study to determine the potential impact of a priori ozone ($O_3$) profiles on TEMPO retrievals in the troposphere. The actual "real-world" TEMPO $O_3$ retrieval algorithm will be non-linear and iterative (Liu et al., 2010). However, the linear approach used in our study has been shown in numerous studies as a good first-order approximation of satellite retrievals of $O_3$ profiles (e.g., Bowman et al., 2002; Worden et al., 2007; Kulawik et al., 2006, 2008; Natraj et al., 2011; Zoogman et al., 2014). The reviewer is also correct that we do not include random retrieval errors ($\varepsilon$) in Eq. (1), however, measurement random-noise error covariance and a priori covariance matrix are included in the calculation of the averaging kernels (AKs) used during this study. Additional text has been added to Sect. 2.2 of the updated manuscript to state these points.

*In retrievals of ozone profiles, an a priori consists of a profile shape and an associated error profile. Because the retrieval of ozone profiles is under-determined (more than one profile shape can be retrieved from the same spectrum), an a priori is used in an Optimal Estimation (OE) based retrieval to constrain the outcome to reasonable values. The a priori profile shape is a reference, and the profile error gives the retrieval the freedom to differ from that reference shape based on the input spectrum to minimize the cost function.*

*In a real retrieval, when either the error on the a priori is set to zero, or the error on the measured/simulated spectrum is too large then the OE retrieval result will reproduce the a priori almost exactly. In this case no information is gained from the spectrum during the retrieval. In other words: the spectrum contains no useful information and the degrees of freedom from the signal (DFS) will be low.*

We agree with the reviewer that the a priori and measurement error are important aspects when calculating retrieval sensitivity. The calculation of the AKs used in this study are described in Zoogman et al. (2017) and the a priori profile and associated error are derived from the TB-Clim

product. Overall, a priori and measurement error terms are taken into account during the calculation of TEMPO AKs applied in Eq. (1).

*The authors seem to come to the conclusion that TEMPO ozone profile retrievals in the troposphere and LMT require an a priori that already matches the general shape of the observations in order for the required accuracy to be obtained in the retrieval. If the a priori already needs to be so close to the shape and magnitude of the outcome of the retrieval, then one could conclude that the TEMPO spectra do not contain sufficient information for the retrieval, or the retrieval is over-constrained.*

*How do the authors see these issues, in light of the need of their conclusion that the a priori needs to be close to the true profile? Please clarify.*

*Another way of looking at it is by looking at Eq 1. If the a priori Xa closely matches the true Xt, then what is 'retrieved' is mainly the a priori, as the second term in the equation falls to zero. It is therefore not surprising that an a priori that more closely matches the true profile will also do well in the simulated retrievals. Those a priori profiles already have the advantage in Eq 1. How does this advantage play out with real retrievals? Is it really necessary to have an a priori so close to the true state of the atmosphere to get a good retrieval? If so, what is the added value of a retrieval in this case?*

The results of this study showing that more accurate a priori trace gas profile assumptions lead to more accurate satellite retrievals in the troposphere are not surprising/novel. The sensitivity of satellite trace gas retrievals to a priori profiles has been clearly stated and demonstrated in numerous studies (e.g., Martin et al., 2002; Luo et al., 2007; Kulawik et al., 2008; Zhang et al., 2010, Bak et al., 2013; many others). These studies, in addition to many others, show that in vertical extents of the atmosphere where satellite sensitivity is low (i.e., middle to lower troposphere for satellites retrieving $O_3$) the retrieved state will be highly dependent on the vertical shape of the a priori. However, our study suggests that the magnitude of the tropospheric-average column $O_3$ abundance will be accurately retrieved by TEMPO regardless of the a prior. This suggests that the magnitude of tropospheric $O_3$ will be largely controlled by the retrieval. The shape of the a priori itself will have a large impact on the shape of the retrieved tropospheric $O_3$ profile and therefore lowermost tropospheric (LMT) $O_3$ magnitudes where satellite sensitivity is low.

The importance of our study is focusing on TEMPO tropospheric $O_3$ retrievals, which due to the system design (geostationary orbit and UV+VIS wavelength retrievals will provide observations with high spatio-temporal variability with increased sensitivity to lower tropospheric $O_3$) will for the first time provide air quality relevant space-borne information. Since TEMPO tropospheric profile $O_3$ data is expected to be assimilated into chemical transport (CTM) and air quality models and LMT data will be used for air quality and event-specific monitoring/research, it is critical to understand methods to improve the quality/accuracy of this retrieved information. Our study demonstrates that to produce TEMPO retrievals of $O_3$ in the LMT with increased accuracy it is necessary to have accurate a priori profile shape assumption. The results from our study also indicate that of all the potential sources of a priori $O_3$ profile data which can be used in satellite retrievals evaluated during this work (climatology data products (e.g., TB-Clim), near-real-time

data assimilation models (e.g., GEOS-5 FP), reanalysis models (e.g., MERRA2), or CTM predictions (e.g., GEOS-Chem)), CTM simulated data result in the most accurate retrievals.

*Textual/other remarks:*

*Line 104: You mention an error margin of the TOLNet measurements of 10% in the lower troposphere and 20% in the upper troposphere. The words 'lower' and 'upper' are not defined in this context, while you use the terms LMT (0-2km) and tropospheric (0-10km). Please be more specific about the applicable altitude ranges of the errors of the TOLNet DIAL lasers.*

We thank the reviewer for this comment as it has led to conversations resulting in an updated and improved statement of TOLNet data uncertainty. The uncertainty of TOLNet $O_3$ retrievals is dependent on numerous factors such as individual instrument specifications, vertical and temporal integration/averaging methods, sampling environment characteristics, etc. Since the TOLNet measurement data used in this study are hourly-averaged and all below 10 km above ground level the updated manuscript has been revised to read: "Uncertainty in TOLNet $O_3$ measurements due to systematic error will be approximately 4-5% for all instruments at almost all altitudes. Precision will vary from 0% to > 20% and is dependent on individual instrument characteristics, time of day, and temporal and vertical averaging (precision typically degrades with height for altitudes above 8-10 km) (Kuang et al., 2013; Sullivan et al., 2015b; Leblanc et al., 2016). Since TOLNet observations used during this study are hourly-averaged and below 10 km above ground-level, overall uncertainty can be assumed to be $\leq 10\%$.".

*In sect 2.2/2.2.1 it would be helpful to have a little more information on the input data. Please elaborate on the setup you use to generate the artificial/simulated TEMPO data (the AK's, the Gain matrices, etc). What other relevant sources of information did you use, like temperature, albedo's, cross sections, solar and viewing angles, reference spectra, etc.*

As stated in the manuscript: "The UV+VIS AKs applied during this study are based on TEMPO retrieval sensitivity studies that play a key role in determining the instrument requirements and verification of the retrieval performance (Zoogman et al., 2017)." In Sect. 7.3 of Zoogman et al. (2017) information is provided about the GEOS-5 meteorological data and GEOS-Chem modeled trace gases and aerosols used to calculate AK values. Viewing geometry, radiance spectra, and weighting functions with respect to aerosols and trace gases are all calculated based on TEMPO specifications as described in Zoogman et al. (2017). Surface albedo values are from the Global Ozone Monitoring Experiment (GOME) albedo database. As mentioned earlier, TB-Clim climatological a prior mean and error covariance matrixes are used in the calculation of TEMPO AKs. To better emphasize the information regarding the AKs that are used during this work that is provided in Zoogman et al. (2017), the following text has been added to Sect. 2.2 of the updated manuscript: "For detailed information about the TEMPO retrieval sensitivity studies, and the input variables, used to derive AKs used during this study see Zoogman et al. (2017).".

*In section 2.2 the authors mention the use adaptation of the SAO retrieval algorithm for TEMPO to do retrievals. But it is not clear to me whether the SAO algorithm played a role in this paper at all. In the second part of 2.2 a simple vector/matrix based formula is used to calculate the simulated retrieved profile. Did the authors use the SAO model for any of the ozone profile*

*retrievals or was it used in the set-up of the kernels? If it was not used, is it then relevant to for this paper?*

The manuscript states "TEMPO will adapt the current SAO OMI UV-only $O_3$ profile algorithm (Liu et al., 2010) to derive $O_3$ profiles from joint UV+VIS measurements based on the optimal estimation technique." to provide an explanation of the TEMPO retrieval algorithm. The SAO algorithm is not used to calculate simulated $O_3$ profile retrievals in this study and are instead approximated using Eq. (1). Please see the above comments which better describe how the AKs used during this study are derived.

*Line 141: In Eq 1, there is a component for the effect of noise. Please explain how you treat the last term in the equation. How does this component affect the retrievals and what are the expectations on its effect on the ranking of the a priori sources used?*

Please see our earlier response that we do not include random retrieval errors ($\varepsilon$) in Eq. (1). This component will add noise to the linear retrievals. Neglecting this will not affect the rankings of the a priori sources. Additional text has been added to Sect. 2.2 of the updated manuscript to clarify this.

*Line 168 and Fig 3: Yellow is a color that is hard to see on a white background. Please use a color with more contrast.*

The yellow line in Fig. 3 has been changed to green in the updated manuscript.

*Line 193: 'due to data constraints'. What kind if data constraints? Is it an issue of lack of sensitivity at the lower troposphere of most existing satellite instruments? Please clarify.*

Both GEOS-5 FP and MERRA2 $O_3$ vertical profiles are primarily driven by the assimilation of Ozone Monitoring Instrument (OMI) and Microwave Limb Sounder (MLS) satellite data. The reviewer is correct in the fact that these satellite products have limited sensitivity in the lower troposphere, and therefore the $O_3$ values from GEOS-5 FP and MERRA2 are most trusted in the upper troposphere and stratosphere. This section of the updated manuscript now reads: "Both GEOS-5 FP and MERRA2 $O_3$ vertical profiles are driven by the assimilation of OMI and Microwave Limb Sounder (MLS) satellite data. Predictions of $O_3$ from these products are most trusted in the upper troposphere and stratosphere due to OMI and MLS having limited sensitivity in the lower troposphere".

*Line 244: In this section you evaluate the straight model output with the TOLNet profiles, outside the context of use as an a priori. The remark that GEOS-Chem is the 'the only potential source of a priori profiles ...' is out of place here. You address the use of the various models as an a priori in sections 3.2.x.*

This has been corrected in the updated manuscript.

*In lines 248 and 249 the authors give a few aspects that may be the reasons why GEOS-Chem compares better to TOLNet than the other models. It would be insightful to the reader to learn which of these aspects contributes the most to the better comparison.*

It would be difficult, and outside the scope of this study, to determine the single reason, out of many, why CTM predictions from GEOS-Chem compare better to $O_3$ observations compared to other data sources evaluated during this study. However, we present the main reasons why one would expect a CTM to predict $O_3$ more accurately compared to GEOS-5 FP, MERRA2, and a climatology product and they are "data-assimilated meteorological fields, comprehensive atmospheric chemistry mechanisms, and state-of-the-art trace gas and aerosol emissions data". We describe in the manuscript that GEOS-5 FP and MERRA2 $O_3$ predictions do not take into account complex atmospheric chemistry routines or emission inventories. Since $O_3$ is a highly reactive trace gas in the troposphere, which has numerous emission sources and production/loss processes, these chemistry routines and emission inventories are necessary to accurately replicate $O_3$ measured in nature.

*Section 3.1.2: In this section the authors make an evaluation of how well the climatology and the models can reproduce the daily variability of the lidar measurements. Please elaborate on the time step/time resolution of the models. Is there a reasonable expectation that the models can actually follow the daily cycle, or are the climatology and model fields spaced to far apart in time?*

In Sect. 2.2.2 the TB-Clim product is described to provide monthly-mean $O_3$ profiles and in Sect. 2.3 the temporal resolution of GEOS-5 FP and MERRA2 are stated to be 3 hours and 10-minute in GEOS-Chem. In Sect. 2.4 it is stated that all measured, modeled, and climatology products are averaged or interpolated to match an hourly temporal resolution for evaluation. The monthly-mean nature of TB-Clim is one of the main reasons why it is unable to replicate the daily and diurnal variability of observed tropospheric $O_3$. The modeled products all have temporal variability of $\leq 3$ hours and therefore have the capability to capture the diurnal variability of $O_3$. However, tropospheric $O_3$ mixing ratios are highly dependent on the diurnal variability of emissions, deposition, and atmospheric chemistry and therefore would be expected to be best replicated from a CTM.

*Please consider enlarging your time series plots.*

This has been done to the best of our ability.

References

Bak, J., Liu, X., Wei, J. C., Pan, L. L., Chance, K., and Kim, J. H.: Improvement of OMI ozone profile retrievals in the upper troposphere and lower stratosphere by the use of a tropopause-based ozone profile climatology, Atmos. Meas. Tech., 6, 2239-2254, https://doi.org/10.5194/amt-6-2239-2013, 2013.

Bowman, K. W., Worden, J., Steck, T., Worden, H. M., Clough, S. and Rodgers, C.: Capturing time and vertical variability of tropospheric ozone: A study using TES nadir retrievals, J. Geophys. Res.-Atmos., 107(D23), 4723, doi:10.1029/2002JD002150, 2002.

Granados-Muñoz, M. J. and Leblanc, T.: Tropospheric ozone seasonal and long-term variability as seen by lidar and surface measurements at the JPL-Table Mountain Facility, California, Atmos. Chem. Phys., 16, 9299-9319, doi:10.5194/acp-16-9299-2016, 2016.

Kuang, S., Newchurch, M. J., Burris, J., and Liu, X.: Ground-based lidar for atmospheric boundary layer ozone measurements, Appl. Opt., 52, 3557-3566, https://doi.org/10.1364/AO.52.003557, 2013.

Kulawik, S. S., Worden, H., Osterman, G., Luo, M., Beer, R., Kinnison, D. E., Bowman, K. W., Worden, J., Eldering, A., Lampel, M., Steck, T., and Rodgers, C. D.: TES atmospheric profile retrieval characterization: An orbit of simulated observations, IEEE T. Geosci. Remote, 44, 1324-1333, 2006.

Kulawik, S. S., Bowman, K.W., Luo, M., Rodgers, C. D., and Jourdain, L.: Impact of nonlinearity on changing the a priori of trace gas profile estimates from the Tropospheric Emission Spectrometer (TES), Atmos. Chem. Phys., 8, 3081–3092, 2008, http://www.atmos-chem-phys.net/8/3081/2008/.

Liu, X., Bhartia, P. K., Chance, K., Spurr, R. J. D., and Kurosu, T. P.: Ozone profile retrievals from the Ozone Monitoring Instrument, Atmos. Chem. Phys., 10, 2521-2537, doi:10.5194/acp-10-2521-2010, 2010.

Martin, R. V., Chance, K., Jacob, D. J., et al.: An improved retrieval of tropospheric nitrogen dioxide from GOME, J. Geophys. Res., 107, 4437, doi:10.1029/2001JD001027, 2002.

Natraj, V., Liu, X., Kulawik, S., Chance, K., Chatfield, R., Edwards, D. P., Eldering, A., Francis, G., Kurosu, T., Pickering, K., Spurr, R., and Worden, H.: Multi-spectral sensitivity studies for the retrieval of tropospheric and lowermost tropospheric ozone from simulated clear-sky GEO-CAPE measurements, Atmos. Environ., 45, 7151-7165, 2011.

Sullivan, J. T., McGee, T. J., Leblanc, T., Sumnicht, G. K., and Twigg, L. W.: Optimization of the GSFC TROPOZ DIAL retrieval using synthetic lidar returns and ozonesondes − Part 1: Algorithm validation, Atmos. Meas. Tech., 8, 4133-4143, doi:10.5194/amt-8-4133-2015, 2015b.

Worden, J., Liu, X., Bowman, K., Chance, K., Beer, R., Eldering, A., Gunson, M., andWorden, H. M.: Improved tropospheric ozone profile retrievals using OMI and TES radiances, Geophys. Res. Lett., 34, L01809, doi:10.1029/2006GL027806, 2007.

Zhang, L., Jacob, D. J., Liu, X., Logan, J. A., Chance, K., Eldering, A., and Bojkov, B. R.: Intercomparison methods for satellite measurements of atmospheric composition: application to tropospheric ozone from TES and OMI, Atmos. Chem. Phys., 10, 4725-4739, https://doi.org/10.5194/acp-10-4725-2010, 2010.

Zoogman, P., Jacob, D. J., Chance, K., Liu, X., Lin, M., Fiore, A., and Travis, K.: Monitoring high-ozone events in the US Intermountain West using TEMPO geostationary satellite

observations, Atmos. Chem. Phys., 14, 6261-6271, https://doi.org/10.5194/acp-14-6261-2014, 2014.

Zoogman, P., Liu, X., Suleiman, R., Pennington, W., Flittner, D., Al-Saadi, J., Hilton, B., Nicks, D., Newchurch, M., Carr, J., Janz, S., Andraschko, M., Arola, A., Baker, B., Canova, B., Miller, C. C., Cohen, R., Davis, J., Dussault, M., Edwards, D., Fishman, J., Ghulam, A., Abad, G. G., Grutter, M., Herman, J., Houck, J., Jacob, D., Joiner, J., Kerridge, B., Kim, J., Krotkov, N., Lamsal, L., Li, C., Lindfors, A., Martin, R., McElroy, C., McLinden, C., Natraj, V., Neil, D., Nowlan, C., O'Sullivan, E., Palmer, P., Pierce, R., Pippin, M., Saiz-Lopez, A., Spurr, R., Szykman, J., Torres, O., Veefkind, J., Veihelmann, B., Wang, H., Wang, J., and Chance, K.: Tropospheric emissions: Monitoring of pollution (TEMPO), J. Quant. Spectrosc. Ra., 186, 17-39, https://doi.org/10.1016/j.jqsrt.2016.05.008, 2017.

---

## Author Response (AR1)

**Response to Reviewer #1 Comments**

We thank the reviewer for their helpful comments. We have incorporated as many of the reviewers' suggestions as possible into the revised manuscript. All reviewer comments are in italics and the author's responses are in standard font.

*The article delivers on its goal of evaluating potential sources of a priori ozone profile information for use in retrievals from TEMPO measurements over North America. The accomplishment is well-summarized by the first sentence of the last paragraph: "This study is a first step in determining what source of a priori vertical O3 profiles should be applied to best enhance the ability of TEMPO to retrieve tropospheric and LMT column O3 in North America."*

*The retrievals envisioned in the article fall into the best-estimate-for-today category of retrieval approaches. That is, they seek to bring in as much information from climatologies or models or other sources as they can into the final near-real-time product. Such approaches may not be well-suited for climate change studies as it can become difficult to unravel the sources of any trends from the influences of the measurements versus the influence of the varying a priori profiles. Even with the averaging kernels and a priori profiles provided for each retrieval, assimilation applications of the data will be more complicated too. Do the developers envision that the models will use these retrievals as input to influence the forecasts?*

The reviewer brings up an important point. The tropospheric ozone ($O_3$) retrieval algorithm for TEMPO is still under development and testing, and therefore the purpose of this study is to determine the general impact of different sources of a priori profiles (climatology products and time-specific (non-climatological) near-real-time (NRT) data assimilation, reanalysis, and chemical transport model (CTM) data) on TEMPO tropospheric and lowermost tropospheric (LMT) $O_3$ column retrievals. As the reviewer identifies, implementing time-specific NRT daily/hourly predictions from CTM or air quality models as the a prior in tropospheric $O_3$ retrievals from TEMPO is best suited when using this output to study topics such as air quality or event-based processes (e.g., air quality exceedances, wildfires, stratospheric intrusions, pollution transport, etc.). Using an a priori from model-predicted NRT daily/hourly information will in fact impact the error/uncertainties and trends of retrieved tropospheric $O_3$ from TEMPO and the final algorithm will likely use an hourly-resolved monthly mean climatology based on model outputs. Based on the results of this study, follow-on research to this manuscript is currently being conducted to develop different CTM-simulated $O_3$ climatology products and test them in the tropospheric $O_3$ retrieval algorithm. It also important to note that the retrieved vertical $O_3$ profiles retrieved from TEMPO can easily be recalculated offline, following methods similar to our work, by data users who want to use a new/different source of a priori.

A major application of TEMPO products is envisioned to be the assimilation of the $O_3$ data (and other chemical constituents) into CTM and air quality models to improve retrospective analysis and forecasts of air quality and tropospheric chemical composition. The standard product of

TEMPO O$_3$ retrievals, or recalculated profiles using a different a priori following the methods of this study, can be assimilated into CTM or air quality models.

To better emphasize these important points, additional text has been added to the updated manuscript, primarily in the conclusions section: "The results of this study clearly demonstrate that using simulated time-specific (non-climatological) O$_3$ profile data will improve near-surface TEMPO O$_3$ retrievals, however, implementing NRT daily/hourly predictions from CTM or air quality models as the a prior is best suited for using TEMPO data to study topics such as air quality or event-based processes (e.g., air quality exceedances, wildfires, stratospheric intrusions, pollution transport, etc.). Applying time-specific daily/hourly predictions from CTM or air quality models as the a priori will impact errors/uncertainties and long-term trends in tropospheric O$_3$ retrievals from TEMPO and these impacts would be difficult to separate from actually retrieved information. Therefore, the standard TEMPO O$_3$ profile algorithm will need to use an hourly-resolved monthly mean climatology and follow-on studies to this manuscript are currently being conducted to develop different CTM-simulated O$_3$ climatology products and test them in the retrieval algorithm. It is important to note that TEMPO data users can easily apply the output from the standard retrieval (e.g., original a priori O$_3$ profile, retrieved O$_3$ profile, and AKs) and recalculate the tropospheric O$_3$ vertical profiles using a new/different source of a priori following the methods of this study. This will allow data users to apply a priori profiles they believe will result in the most accurate/representative tropospheric and LMT O$_3$ magnitudes from TEMPO without having to rerun the computationally-expensive SAO retrieval algorithm.". Text has also been added to the abstract: "The application of time-specific (non-climatological) hourly/daily model predictions as the a priori profile in TEMPO O$_3$ retrievals will be best suited when applying this data to study air quality or event-based processes as the standard retrieval algorithm will still need to use a climatology product. Follow-on studies to this work are currently being conducted to investigate the application of different CTM-predicted O$_3$ climatology products in the standard TEMPO retrieval algorithm. Finally, similar methods to those used in this study can be easily applied by TEMPO data users to recalculate tropospheric O$_3$ profiles provided from the standard retrieval using a different source of a priori." and Sect. 2.3: "Due to numerous reasons the standard TEMPO O$_3$ profile algorithm will need to apply an hourly-resolved monthly mean climatology, however, we evaluated time-specific model data here as TEMPO data users can simply apply the outputs from the standard retrieval to recalculate the tropospheric O$_3$ vertical profiles using a different source of a priori.".

*A key performance index for the study is the ability of the retrieved profiles to identify high ozone levels in the lowermost troposphere (LMT 0-2km). With this in mind, Tables 4 and 5 should give correlations so that the readers can better compare the performance of the a priori profiles alone, provided in the earlier tables, to the performance of the retrieved profiles.*

We agree with the reviewer and these correlation values have been added for tropospheric and LMT column $O_3$. Some minor text has also been added to the updated manuscript to explain these results in Sect. 3.2.1 and 3.2.2.

*I was surprised that the article does not include a discussion of the effects of surface reflectivity (and knowledge of the surface reflectivity and surface pressure) on the lower layer information content. What ground reflectivity was assumed in the clear sky retrievals? How will seasonal variability, especially snow cover, be addressed in the algorithm? A future study could also consider the use of clear versus cloudy or partially cloudy (with cloud height and cloud fraction information from the measurements) results for adjacent pixels to try to identify the below cloud columns better (or even to apply some version of cloud slicing).*

We agree with the reviewer that near-surface $O_3$ retrievals from ultraviolet + visible (UV+VIS) wavelengths are sensitive to surface reflectance/albedo (primarily in the VIS). TEMPO retrieval sensitivity studies which produced the averaging kernels (AK) used during this study (see Zoogman et al. (2017)) applied surface albedo values from the Global Ozone Monitoring Experiment (GOME) albedo database and surface pressure was taken from the GEOS-5 meteorological model. The GOME database provides a monthly mean surface albedo climatology at a spatial resolution of $1° \times 1°$ for multiple wavelengths (from 335 to 772 nm) which were interpolated/extrapolated to match TEMPO retrieved wavelengths. The spatio-temporal variability of snow-cover is taken into account when producing TEMPO AKs, but for this study, which is focused on summer-months, will not have any impact on the results. In the actual TEMPO retrieval, surface albedo will be retrieved as a first-order polynomial in the UV following Liu et al. (2005, 2010) and a new climatology of visible surface albedo spectra has been developed for fitting surface albedo spectra in the visible using multiple parameters (Zoogman et al., 2016). Surface albedo is typically well retrieved from this algorithm and its effect on the retrieval sensitivity/information content is taken into account.

We also agree with the reviewer that a future study focused on the impact of clouds (e.g., fraction height, etc.) would be interesting.

Discussing all the input variables and data sources used in the production of AKs used during this work is outside the scope of this manuscript. However, all this information is presented in Zoogman et al. (2017) and therefore the following text has been added to Sect. 2.2.1 of the updated manuscript: "For detailed information about the TEMPO retrieval sensitivity studies, and the input variables, used to derive AKs applied during this study see Zoogman et al. (2017).".

*Editorial erratum*

*Table 3 does not contain a listed section for JPL TMF results.*

Table 3 does not have a listed section for JPL TMF results as this table presents the statistics of the comparison of the diurnal time-series of hourly-averaged tropospheric and LMT $O_3$ from the climatology and models to observations. No hourly-averaged lidar observations were available from the JPL TMF system for diurnal time-series evaluation as stated in Sect. 2.1 of the manuscript: "During the summer of 2014, the JPL TMF lidar only conducted measurements during the nighttime hours and therefore will only be used for daily-averaged comparisons to TB-Clim and model predictions".

To better explain this, the updated manuscript in Sect. 2.4 now reads: "Due to the hours of operation, the evaluation at the JPL TMF lidar location was not conducted for hourly-averages and is only applied for summer- and daily-averages.".

**Response to Reviewer #2 Comments**

We thank the reviewer for their helpful comments. We have incorporated as many of the reviewers' suggestions as possible into the revised manuscript. All reviewer comments are in italics and the author's responses are in standard font.

*In this paper, the authors compare ozone profile data from three TOLNet stations across the USA with A) an ozone climatology and ozone data from various transport models, and B) a simulated retrieval result where the climatology and models are used as a priori values.*

*The authors use a formula from the book by Rodgers (2000) to linearise the calculations of the effect of the a priori on a potential retrieval. While Rodgers uses the formula in Chap 3 and Chap 10 of his book, using this formula to make a selection on a preferred a priori brushes over the potential issues you often get with real satellite data.*

*The first question that comes to mind is: how representative are these simulated retrievals for real world situations. Or is Eq 1 limited to be used for an error / sensitivity study? My impression is that the error component is not used in the paper. Please give your reasons for using this method.*

The reviewer is correct in the fact that the application of Eq. (1) in our study is representative of a sensitivity study to determine the potential impact of a priori ozone ($O_3$) profiles on TEMPO retrievals in the troposphere. The actual "real-world" TEMPO $O_3$ retrieval algorithm will be non-linear and iterative (Liu et al., 2010). However, the linear approach used in our study has been shown in numerous studies as a good first-order approximation of satellite retrievals of $O_3$ profiles (e.g., Bowman et al., 2002; Worden et al., 2007; Kulawik et al., 2006, 2008; Natraj et al., 2011; Zoogman et al., 2014). The reviewer is also correct that we do not include random retrieval errors ($\varepsilon$) in Eq. (1), however, measurement random-noise error covariance and a priori covariance matrix are included in the calculation of the averaging kernels (AKs) used during this study.

Additional text has been added to Sect. 2.2 of the updated manuscript to state these points: "This linear estimation approach is a good first-order approximation of non-linear satellite retrievals and has been used in numerous research studies (e.g., Bowman et al., 2002; Worden et al., 2007;

Kulawik et al., 2006, 2008; Natraj et al., 2011; Zoogman et al., 2014)." and "The last term on the right represents the retrieval precision. During this study, no measurement noise/error is taken into account. The error component adds measurement noise to the linear retrievals, however, neglecting this term does not affect the inter-comparison of the impact of individual a priori sources on TEMPO retrieved tropospheric $O_3$.".

*In retrievals of ozone profiles, an a priori consists of a profile shape and an associated error profile. Because the retrieval of ozone profiles is under-determined (more than one profile shape can be retrieved from the same spectrum), an a priori is used in an Optimal Estimation (OE) based retrieval to constrain the outcome to reasonable values. The a priori profile shape is a reference, and the profile error gives the retrieval the freedom to differ from that reference shape based on the input spectrum to minimize the cost function.*

*In a real retrieval, when either the error on the a priori is set to zero, or the error on the measured/simulated spectrum is too large then the OE retrieval result will reproduce the a priori almost exactly. In this case no information is gained from the spectrum during the retrieval. In other words: the spectrum contains no useful information and the degrees of freedom from the signal (DFS) will be low.*

We agree with the reviewer that the a priori and measurement error are important aspects when calculating retrieval sensitivity. The calculation of the AKs used in this study are described in Zoogman et al. (2017) and the a priori profile mean and associated error are derived from the TB-Clim product. Overall, a priori and measurement error terms are taken into account during the calculation of TEMPO AKs applied in Eq. (1).

Minor text has been added to the sentence in Sect. 2.2.1 addressing this point: "The production of these AKs involved: 1) radiative transfer model simulations of TEMPO radiance spectra and weighting functions, 2) retrieval AKs and errors constrained by the TB-Clim a priori mean and error covariance matrix, and 3) measurement errors estimated using the TEMPO signal to noise ratio model.".

*The authors seem to come to the conclusion that TEMPO ozone profile retrievals in the troposphere and LMT require an a priori that already matches the general shape of the observations in order for the required accuracy to be obtained in the retrieval. If the a priori already needs to be so close to the shape and magnitude of the outcome of the retrieval, then one could conclude that the TEMPO spectra do not contain sufficient information for the retrieval, or the retrieval is over-constrained.*

*How do the authors see these issues, in light of the need of their conclusion that the a priori needs to be close to the true profile? Please clarify.*

*Another way of looking at it is by looking at Eq 1. If the a priori Xa closely matches the true Xt, then what is 'retrieved' is mainly the a priori, as the second term in the equation falls to zero. It is therefore not surprising that an a priori that more closely matches the true profile will also do well in the simulated retrievals. Those a priori profiles already have the advantage in Eq 1. How does this advantage play out with real retrievals? Is it really necessary to have an a priori so close*

*to the true state of the atmosphere to get a good retrieval? If so, what is the added value of a retrieval in this case?*

The results of this study showing that more accurate a priori trace gas profile assumptions lead to more accurate satellite retrievals in the troposphere are not surprising/novel. The sensitivity of satellite trace gas retrievals to a priori profiles has been clearly stated and demonstrated in numerous studies (e.g., Martin et al., 2002; Luo et al., 2007; Kulawik et al., 2008; Zhang et al., 2010, Bak et al., 2013). These studies, in addition to many others, show that in vertical extents of the atmosphere where satellite sensitivity is low (i.e., middle to lower troposphere for satellites retrieving $O_3$) the retrieved state will be dependent on the vertical shape of the a priori. Overall, our study shows that the magnitude of the tropospheric-average column $O_3$ abundance will be accurately retrieved by TEMPO regardless of the a prior. This suggests that the magnitude of tropospheric $O_3$ will be largely controlled by the retrieval. The shape of the a priori itself will have a large impact on the shape of the retrieved tropospheric $O_3$ profile and therefore lowermost tropospheric (LMT) $O_3$ magnitudes where satellite sensitivity is low.

The importance of our study is focusing on TEMPO tropospheric $O_3$ retrievals, which due to the system design (geostationary orbit and UV+VIS wavelength retrievals will provide observations with high spatio-temporal variability with increased sensitivity to lower tropospheric $O_3$) will for the first time provide air quality relevant space-borne information. Since TEMPO tropospheric profile $O_3$ data is expected to be assimilated into chemical transport (CTM) and air quality models and LMT data will be used for air quality and event-specific monitoring/research, it is critical to understand methods to improve the quality/accuracy of this retrieved information. Our study demonstrates that to produce TEMPO retrievals of $O_3$ in the LMT with increased accuracy it is necessary to have accurate a priori profile shape assumption. The results from our study also indicate that of all the potential sources of a priori $O_3$ profile data which can be used in satellite retrievals evaluated during this work (climatology data products (e.g., TB-Clim), near-real-time data assimilation models (e.g., GEOS-5 FP), reanalysis models (e.g., MERRA2), or CTM predictions (e.g., GEOS-Chem)), time-specific CTM simulated data result in the most accurate retrievals.

To better emphasize these points text has been added to Sect. 3.2.3: "While the magnitude of the tropospheric $O_3$ column will be largely controlled by the retrieval, the shape of the a priori profile itself will have an impact on the shape of the retrieved tropospheric $O_3$ profile, and therefore the LMT $O_3$ magnitudes where satellite sensitivity is low." and conclusions section of the updated manuscript: "In general, the magnitude of the tropospheric $O_3$ column from TEMPO will be largely controlled by the retrieval and the shape of the a priori profile will have a noticeable impact on the shape of the retrieved tropospheric $O_3$ profile, and therefore the LMT $O_3$ magnitudes where satellite sensitivity is low.".

*Textual/other remarks:*

*Line 104: You mention an error margin of the TOLNet measurements of 10% in the lower troposphere and 20% in the upper troposphere. The words 'lower' and 'upper' are not defined in this context, while you use the terms LMT (0-2km) and tropospheric (0-10km). Please be more specific about the applicable altitude ranges of the errors of the TOLNet DIAL lasers.*

We thank the reviewer for this comment as it has led to conversations resulting in an updated and improved statement of TOLNet data uncertainty. The uncertainty of TOLNet $O_3$ retrievals is dependent on numerous factors such as individual instrument specifications, vertical and temporal integration/averaging methods, sampling environment characteristics, etc. Since the TOLNet measurement data used in this study are hourly-averaged and all generally sampled below 10 km above ground level the updated manuscript has been revised to read: "Uncertainty in TOLNet $O_3$ measurements due to systematic error are approximately 4-5% for all instruments at all altitudes. Precision will vary from 0% to > 20% and is dependent on individual instrument characteristics, time of day, and temporal and vertical averaging (precision typically degrades with height for altitudes above 8-10 km) (Kuang et al., 2013; Sullivan et al., 2015b; Leblanc et al., 2016). Since TOLNet observations used during this study are hourly-averaged and typically below 10 km agl, overall uncertainty can be assumed to be ≤ 10%.".

*In sect 2.2/2.2.1 it would be helpful to have a little more information on the input data. Please elaborate on the setup you use to generate the artificial/simulated TEMPO data (the AK's, the Gain matrices, etc). What other relevant sources of information did you use, like temperature, albedo's, cross sections, solar and viewing angles, reference spectra, etc.*

As stated in the manuscript: "The UV+VIS AKs applied during this study are based on TEMPO retrieval sensitivity studies that play a key role in determining the instrument requirements and verification of the retrieval performance (Zoogman et al., 2017)." In Sect. 7.3 of Zoogman et al. (2017) information is provided about the GEOS-5 meteorological data and GEOS-Chem modeled trace gases and aerosols used to calculate AK values. Viewing geometry, radiance spectra, and weighting functions with respect to aerosols and trace gases are all calculated based on TEMPO specifications as described in Zoogman et al. (2017). Surface albedo values are from the Global Ozone Monitoring Experiment (GOME) albedo database. As mentioned earlier, TB-Clim climatological a prior mean and error covariance matrixes are used in the calculation of TEMPO AKs.

To better emphasize the information regarding the AKs that are used during this work that is provided in Zoogman et al. (2017), the following text has been added to Sect. 2.2 of the updated manuscript: "For detailed information about the TEMPO retrieval sensitivity studies, and the input variables, used to derive AKs applied during this study see Zoogman et al. (2017).".

*In section 2.2 the authors mention the use adaptation of the SAO retrieval algorithm for TEMPO to do retrievals. But it is not clear to me whether the SAO algorithm played a role in this paper at all. In the second part of 2.2 a simple vector/matrix based formula is used to calculate the simulated retrieved profile. Did the authors use the SAO model for any of the ozone profile retrievals or was it used in the set-up of the kernels? If it was not used, is it then relevant to for this paper?*

The manuscript states "TEMPO will adapt the current SAO OMI UV-only $O_3$ profile algorithm (Liu et al., 2010) to derive $O_3$ profiles from joint UV+VIS measurements based on the optimal estimation technique." to provide an explanation of the TEMPO retrieval algorithm. The SAO algorithm is not used to calculate simulated $O_3$ profile retrievals in this study and are instead

approximated using Eq. (1). Please see the above comments which better describe how the AKs used during this study are derived.

*Line 141: In Eq 1, there is a component for the effect of noise. Please explain how you treat the last term in the equation. How does this component affect the retrievals and what are the expectations on its effect on the ranking of the a priori sources used?*

Please see our earlier response that we do not include random retrieval errors (ε) in Eq. (1). This component will add noise to the linear retrievals. Neglecting this will not affect the rankings of the a priori sources.

Additional text has been added to Sect. 2.2 of the updated manuscript to clarify this: "The last term on the right represents the retrieval precision. During this study, no measurement noise/error is taken into account. The error component adds measurement noise to the linear retrievals, however, neglecting this term does not affect the inter-comparison of the impact of individual a priori sources on TEMPO retrieved tropospheric $O_3$.".

*Line 168 and Fig 3: Yellow is a color that is hard to see on a white background. Please use a color with more contrast.*

The yellow line in Fig. 3 has been changed to green in the updated manuscript.

*Line 193: 'due to data constraints'. What kind if data constraints? Is it an issue of lack of sensitivity at the lower troposphere of most existing satellite instruments? Please clarify.*

Both GEOS-5 FP and MERRA2 $O_3$ vertical profiles are primarily driven by the assimilation of Ozone Monitoring Instrument (OMI) and Microwave Limb Sounder (MLS) satellite data. The reviewer is correct in the fact that these satellite products have limited sensitivity in the lower troposphere, and therefore the $O_3$ values from GEOS-5 FP and MERRA2 are most trusted in the upper troposphere and stratosphere.

This section of the updated manuscript now reads: "Both GEOS-5 FP and MERRA2 $O_3$ vertical profiles are driven by the assimilation of OMI and Microwave Limb Sounder (MLS) satellite data. Predictions of $O_3$ from these products are most trusted in the upper troposphere and stratosphere due to OMI and MLS having limited sensitivity in the lower troposphere (e.g., Wargan et al., 2015; Ott et al., 2016).".

*Line 244: In this section you evaluate the straight model output with the TOLNet profiles, outside the context of use as an a priori. The remark that GEOS-Chem is the 'the only potential source of a priori profiles ...' is out of place here. You address the use of the various models as an a priori in sections 3.2.x.*

This has been corrected in the updated manuscript.

*In lines 248 and 249 the authors give a few aspects that may be the reasons why GEOS-Chem compares better to TOLNet than the other models. It would be insightful to the reader to learn which of these aspects contributes the most to the better comparison.*

It would be difficult, and outside the scope of this study, to determine the single reason, out of many, why CTM predictions from GEOS-Chem compare better to $O_3$ observations compared to other data sources evaluated during this study. However, we present the main reasons why one would expect a CTM to predict $O_3$ more accurately compared to GEOS-5 FP, MERRA2, and a climatology product and they are "data-assimilated meteorological fields, comprehensive atmospheric chemistry mechanisms, and state-of-the-art trace gas and aerosol emissions data". We describe in the manuscript that GEOS-5 FP and MERRA2 $O_3$ predictions do not take into account complex atmospheric chemistry routines or emission inventories. Since $O_3$ is a highly reactive trace gas in the troposphere, which has numerous emission sources and production/loss processes, these chemistry routines and emission inventories are necessary to accurately replicate $O_3$ measured in nature.

*Section 3.1.2: In this section the authors make an evaluation of how well the climatology and the models can reproduce the daily variability of the lidar measurements. Please elaborate on the time step/time resolution of the models. Is there a reasonable expectation that the models can actually follow the daily cycle, or are the climatology and model fields spaced to far apart in time?*

In Sect. 2.2.2 of the manuscript the TB-Clim product is described to provide monthly-mean $O_3$ profiles and in Sect. 2.3 the GEOS-5 FP and MERRA2 data are available as 3 hour-averages and 10-minutes in GEOS-Chem. In Sect. 2.4 it is stated that all measured, modeled, and climatology products are averaged or interpolated to an hourly temporal resolution for evaluation. The monthly-mean nature of TB-Clim is one of the main reasons why it is unable to replicate the daily and diurnal variability of observed tropospheric $O_3$. However, the GEOS-5 (used to produce GEOS-5 FP and MERRA2) and GEOS-Chem models both have transport timesteps of $\leq 10$ minutes and therefore have the capability to capture the diurnal variability of $O_3$. However, tropospheric $O_3$ mixing ratios are highly dependent on the diurnal variability of emissions, deposition, and atmospheric chemistry and therefore would be expected to be best replicated from a CTM (i.e., GEOS-Chem) as these processes are not taken into account in GEOS-4 FP and MERRA2.

*Please consider enlarging your time series plots.*

This has been done to the best of our ability.

[revised manuscript text omitted]
)analysis has demonstrated the high accuracy of TOLNet $O_3$ retrievals with errors typically estimated to be around ±10% in the lower troposphere and ±20% in the upper troposphere (Kuang et al., 2013; Sullivan et al., 2015b; Granados-Muñoz and Leblanc et al., 2016). Since TOLNet observations used during this study are hourly-averaged and typically below 10 km agl, overall uncertainty can be assumed to be ≤ 10%. TOLNet data will bewere applied in this study to evaluate the TB-Clim and model-predicted profiles which could potentially be used as TEMPO a priori information. Furthermore, theoretical TEMPO $O_3$ retrievals in the troposphere and LMT were calculated using the climatology/model profiles as a priori with TOLNet data representing the "true" atmospheric $O_3$ profiles (see Sect. 2.2).

[revised manuscript text omitted]

**2.2.1 TEMPO averaging kernels**

The UV+VIS AKs applied during this study are based on TEMPO retrieval sensitivity studies that play a key role in determining the instrument requirements and verification of the retrieval performance (Zoogman et al., 2017). The production of these AKs involved: 1) radiative transfer model simulations of TEMPO radiance spectra and weighting functions, 2) retrieval AKs and errors constrained by the TB-Clim a priori mean and error covariance matrix, and 3) measurement errors estimated using the TEMPO signal to noise ratio model. To represent TEMPO hourly measurements throughout the year, the retrieval sensitivity calculation was performed hourly for 12 days (15[th] day of each month) over the TEMPO domain at a spatial resolution of 2.0°×2.5° (latitude × longitude) using hourly GEOS-Chem model fields. For detailed information about the TEMPO retrieval sensitivity studies, and the input variables, used to derive AKs applied during this study see Zoogman et al. (2017). 
[revised manuscript text omitted]
. what source of a priori vertical $O_3$ profiles should be applied to best enhance the ability of TEMPO to retrieve tropospheric and LMT column $O_3$ in North America. It The results demonstrates that model simulations, in particular those from a CTM, improve TEMPO tropospheric $O_3$ retrievals over climatological products such as TB-Clim when applied as the a priori data. However, there are instances where CTM predictions do did not improve TEMPO retrieved values compared to the TB-Clim data. Furthermore, out of the 59 total days of TOLNet observations analyzed during this study, LMT column $X_r$ values using GEOS-Chem a priori profiles show biases greater than the TEMPO 10 ppb accuracy requirement for ~15% of the days. It should be noted that this number of LMT column $X_r$ error exceedances is the least compared to when using all the sources of a priori and greater than a factor of 2 smaller than when applying TB-Clim a priori. The main reason for the majority of error exceedances is because the a priori profiles cannot do not capture the dynamic vertical $O_3$ profile observed by the TOLNet lidars. Therefore, further work is needed to identify the source of a priori $O_3$ profiles for use in TEMPO $O_3$ retrievals which can best capture the shape of tropospheric $O_3$ profiles in North America.

[revised manuscript text omitted]